# A conserved odorant binding protein is required for essential amino acid detection in Drosophila

Karen Rihani[1], Stéphane Fraichard[1], Isabelle Chauvel[1], Nicolas Poirier[1], Thomas Delompré[1], Fabrice Neiers[1], Teiichi Tanimura[2,3], Jean-François Ferveur [1]* & Loïc Briand [1]*

Animals need to detect in the food essential amino acids that they cannot synthesize. We found that the odorant binding protein OBP19b, which is highly expressed in *Drosophila melanogaster* taste sensilla, is necessary for the detection of several amino acids including the essential L-phenylalanine. The recombinant OBP19b protein was produced and characterized for its binding properties: it stereoselectively binds to several amino acids. Using a feeding-choice assay, we found that OBP19b is necessary for detecting L-phenylalanine and L-glutamine, but not L-alanine or D-phenylalanine. We mapped the cells expressing OBP19b and compared the electrophysiological responses of a single taste sensillum to several amino acids: OBP19b mutant flies showed a reduced response compared to control flies when tested to preferred amino acids, but not to the other ones. OBP19b is well conserved in phylogenetically distant species suggesting that this protein is necessary for detection of specific amino acids in insects.

[1] AgroSup Dijon, CNRS, INRA, Université de Bourgogne-Franche Comté, Centre des Sciences du Goût et de l'Alimentation, 21000 Dijon, France. [2] Division of Biological Science, Graduate School of Science, Nagoya University, Furo, Chikusa, Aichi 464-8602, Japan. [3] Department of Genetics, Leibniz Institute for Neurobiology (LIN), Brenneckestrasse 6, 39118 Magdeburg, Germany. *email: jean-francois.ferveur@u-bourgogne.fr; loic.briand@inra.fr

Animals can survive if they are able to detect nutritious food sources and avoid toxic ones. Among available nutrients, amino acids are necessary for the growth, development, reproduction, and survival of both vertebrates and invertebrates. Amino acids also play a role in neuronal signalization, protein phosphorylation and gene regulation[1,2]. Some amino acids are endogenously synthesized by animals, while others—essential amino acids—cannot be synthesized and are found only in the diet. Amino acid sensing depends on the detection of intracellular amino acid levels through two signal transduction pathways involved in cell growth and metabolism. In eukaryotes, the general control non-derepressible 2 pathway detects the absence of amino acids, whereas the Target of Rapamycin kinase pathway process particular amino acids[3,4]. In insects, amino acid preference depends on the internal nutritional state: an amino acid—poor diet can affect larval development, egg production and lifespan[5,6], while amino acid-deprived adults show increased amino acid consumption[7]. Amino acid preference can also change during development[8] and after mating[9].

During evolution, animals have likely developed the gustatory ability to detect amino acids in their environment with their peripheral chemosensory system[10]. Many amino acids attract rodents and elicit a umami or sweet taste response in humans[10]. In mammals, amino acid detection involves a heterodimeric complex of TAS1R1 and TAS1R3 G protein-coupled receptors expressed in the taste receptor cells of taste buds located on taste papillae in the oral cavity[11]. In contrast, insects, which do not possess homologous amino acid taste receptors, use alternative mechanisms. *Drosophila melanogaster* likely detects amino acids with the ionotropic chemosensory receptor Ir76b[12,13] expressed in the larval pharyngeal and external chemosensory organs and the adult tarsal, labellum, and pharyngeal taste neurons[13,14].

Only a few perireceptor events underlying the detection of food compounds have been molecularly characterized, but none of them pertains to amino acid detection. A well-studied example concerns the perireceptor cascade of events, leading to the detection of the male pheromone *cis*-vaccenyl acetate (cVA). To reach cVA-dedicated receptors, cVA needs to be solubilized and transported through the hydrophilic sensillum lymph[15–17]. This process involves the soluble odorant binding protein (OBP) LUSH (also known as OBP76a), which belongs to a 51-member family. Although the details of this mechanism have been debated[16], the conformational changes in LUSH were shown to mediate neuron activation through OR67d and SNMP proteins. This finding suggests that LUSH does not act simply as a passive cVA carrier[15]. Of the OBPs that were initially described in olfactory sensilla[18] (and named accordingly), some have also been detected in taste sensilla[19–23]. Among the different physiological roles that OBPs have been proposed to play in olfaction[24–27], only their function as lipophilic compound carriers has been demonstrated. Among the very few data available on the putative gustatory role of OBPs, OBP49a was shown to suppress the appetence for sweet compounds through the perception of bitter tastants[28]. Recently, the Drosophila OBP59a was found to play a role in hygroreception[29]. Here, we took a multifaceted approach to characterize the binding properties and the expression and function of the taste-expressed OBP19b. We found that *D. melanogaster* adults need to express this soluble protein in their taste organs to detect specific amino acids, including the essential L-phenylalanine.

## Results

### Abundant expression of OBP19b in gustatory organs.

To screen OBPs potentially involved in taste detection in *D. melanogaster* adults, we measured the transcription level of several

**Table 1 Expression levels of OBP transcripts measured by RT-qPCR in head olfactory and taste appendages of adult male *Drosophila melanogaster*.**

|  | Olfactory appendages | Taste appendages |
|---|---|---|
| OBP19b | 13 | 403 |
| OBP19d | 239 | 10 |
| OBP28a | 355 | 2.3 |
| OBP56e | 0.4 | 2.5 |
| OBP57b | 3.0 | 5.5 |
| OBP83a | 561 | 0.2 |
| OBP83b | 474 | 0.2 |

Data are given in arbitrary units. The head (without appendages), the thorax and the abdomen were used for reference ($N = 3$)

OBPs previously reported to be expressed in taste appendages[19,28,30]. We compared transcription expression levels in the proboscis and the olfactory appendages of the head (antenna + maxillary palps) with those found in other parts of the body (minus the three removed head appendages). Three OBPs (OBP19b, OBP56e, and OBP57b) showed a higher transcription level in the proboscis than in the other tissues (Table 1). We focused on OBP19b, which showed a remarkably high-expression level in the proboscis compared to head olfactory organs.

**Heterologous expression and purification of OBP19b.** We used methylotropic *Pichia pastoris* yeast to produce large amounts of recombinant OBP19b. The secreted protein was purified using cation-exchange chromatography followed by gel filtration. The purity of the purified OBP19b was checked using sodium dodecyl sulfate–polyacrylamide gel electrophoresis (SDS-PAGE). A single band, corresponding to the expected molecular mass, was observed migrating at 15 kDa (Fig. 1a). Using Matrix Assisted Laser Desorption Ionisation-Time of Flight (MALDI-ToF) mass spectrometry, the OBP19b protein mass was found to be 15,264.5 Da, in agreement with the predicted mass of the mature protein (Fig. 1b). The protein folding was determined using circular dichroism spectroscopy, a biophysical technique for monitoring secondary structures of a protein in solution. The far-UV circular dichroism spectrum of OBP19b (Fig. 1c) revealed a positive peak (193 nm) and two negative peaks (206 and 225 nm), indicating an abundance of alpha helices. The OBP19b secondary structures is in agreement with the expected tertiary structures of insect OBPs with a hydrophobic-binding pocket formed by five or six alpha helices[31–33].

**Ligand-binding properties of OBP19b.** To determine the binding properties of OBP19b, we used the competitive *N*-phenyl-1-naphthylamine (NPN) fluorescence assay. We first measured the ability of OBP19b to reversibly bind NPN. After titration of OBP19b with NPN, a dissociation constant value of $4.1 \pm 0.9 \, \mu M$ was calculated (Supplementary Fig. 1). Then, the affinity of OBP19b for potential ligands was tested with a competitive-binding assay to reveal their respective ligand capacity to displace NPN from the OBP cavity. In agreement with the physiological sensing ability of Drosophila, the taste compounds screened were either tested at a final concentration of $40 \, \mu M$ (Fig. 2a) or 15 mM (Fig. 2b). The highest level of fluorescence displacement was induced by papaverine, berberine, 2-phenylethyl isothiocyanate, D-glucose, D-trehalose, and L-phenylalanine, an essential amino acid for *D. melanogaster* flies[5] (Fig. 2a, b). Given the unexpected effect of L-phenylalanine, we measured the binding ability of OBP19b to all other L-amino acids. In particular, L-glutamine, L-arginine, L-methionine, L-aspartic acid, L-asparagine, and L-serine induced a

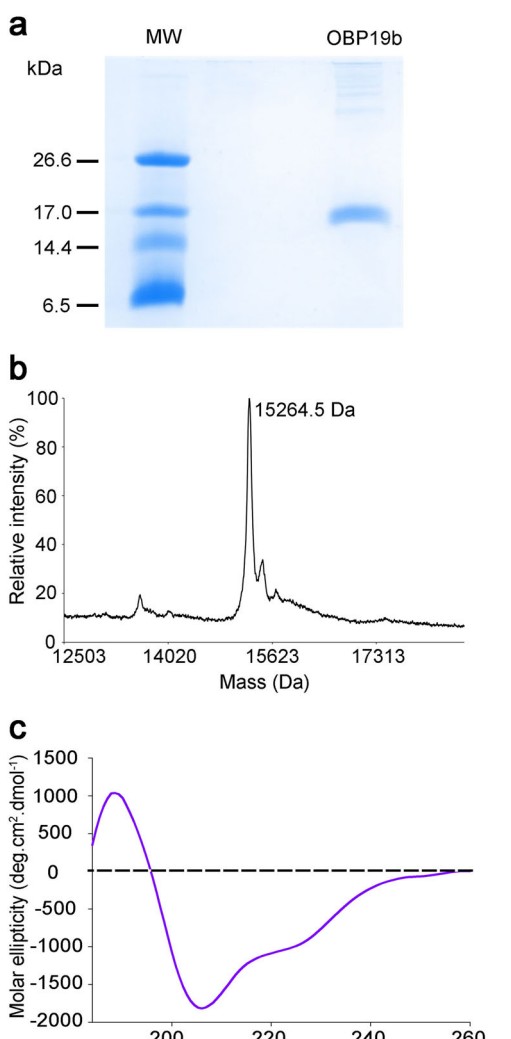

**Fig. 1** Molecular characterization of recombinant OBP19b. **a** SDS-PAGE analysis of purified OBP19b. Proteins were stained using Coomassie Brilliant Blue. The MW lane shows the molecular weight standard 6.5–26.6 kDa (Ultra-Low Range marker, Sigma-Aldrich). **b** MALDI-ToF mass spectrometry analysis of OBP19b. **c** Characterization of OBP19b folding based on circular dichroism spectroscopy. Far-UV circular dichroism spectrum of OBP19b. The protein concentration was 5.0 mg/mL in 50 mM sodium phosphate at pH 7.5.

very significant reduction in the fluorescence intensity of the OBP19b-NPN complex ($p < 0.01$—$p < 0.001$; Fig. 2c). No other L-amino acid induced a significant effect. To check for amino acid-binding specificity, we tested the ability of three D-amino acid stereoisomers (D-phenylalanine, D-glutamine, D-alanine) to displace NPN. The D-isomers induced no significant fluorescence reduction of the OBP19b-NPN complex (Fig. 2c). We also tested the ability of OBP19b to bind three nonproteinogenic amino acids known to affect Drosophila development (L-citrulline, L-ornithine, and L-canavanine)[34,35]. These compounds did not induce a significant effect (Fig. 2c). We further assessed the affinity of amino acids identified as ligands for OBP19b using dose-response measurements (Fig. 2d). The dissociation constant values revealed an affinity in the millimolar range (Supplementary Fig. 2). In the next part of our study, we focused on L-phenylalanine and L-glutamine based on their higher binding affinities for OBP19b (Fig. 2d and Supplementary Fig. 2b). We also used L-alanine and D-phenylalanine as negative controls.

## OBP19b cellular expression

Given the high level of OBP19b transcript in the proboscis, we mapped the cells and sensilla expressing this OBP. We targeted fluorescent transgenes separately in two types of accessory cells (Fig. 3a) surrounding the base of taste sensory neurons found in proboscis sensilla: tormogen cells (using *ASE5*-GFP; Fig. 3b) and thecogen cells (using *nompA*-GFP; Fig. 3c). Simultaneously, for each of these transgenes, we targeted OBP19b expression (with the OBP19b-Gal4 > UAS-mCherry transgenes producing a magenta fluorescent pattern). The fact that OBP19b expression overlapped with that of *nompA*-GFP—but not that of *ASE5*-GFP—at the base of some proboscis sensilla indicates that OBP19b is expressed in the thecogen cells but not in the tormogen cells. Such OBP19b expression in the thecogen cells (Fig. 3e) allowed us to map the sensilla expressing this OBP: nine s-type sensilla (s1 and s5-s12 but not determined in s2-s4) and all i-type sensilla; l-type sensilla showed no OBP19b expression.

## OBP19b is involved in the taste preference of several amino acids

To determine the function of OBP19b in the amino acid feeding preference of adults, we used the CAFE assay (Fig. 4a). This test enabled us to compare, after 6 h, the consumption of two types of food: one containing an amino acid and one without an amino acid. The quantitative difference between the levels of the two foods consumed was used to estimate the preference index (PI).

Previous studies have shown sexually dimorphic feeding preference for amino acids. In addition, different behavioral preferences have been detected between mated and virgin females[13]. We first measured the preference for 15 mM L-phenylalanine in mature female flies. In the control genotype, mated females showed a higher PI than virgin females (Fig. 4b). In contrast, OBP19b-null mutant females (mated or virgin) were indifferent to this amino acid (PI = 0). To rule out a possible interaction induced by the sucrose used in our CAFE assay, sucrose was eliminated from the assay: mated females had the choice between pure 15 mM L-phenylalanine or the water used to dissolve the amino acid (Fig. 4c). In this test, the control mated females showed a positive PI (+0.3) to the pure amino acid, whereas mutant females were indifferent. The preference for 15 mM L-glutamine, L-alanine was also measured in mated female flies. Control females showed a clear preference for L-glutamine (PI = +0.4), while mutant females were indifferent to this amino acid. Moreover, control and mutant females showed similar PIs to L-alanine (Supplementary Fig. 3).

Then, we tested the response of mature males by giving them a choice between 1 mM sucrose and amino acid-containing solutions (1 mM sucrose + 15 mM amino acid, which was either L-phenylalanine, L-glutamine, L-alanine, or D-phenylalanine) (Fig. 4d). Control males showed a clear preference for the three tested L-amino acids (median PI range of +0.2 to +0.4), while mutant males were indifferent to L-phenylalanine and L-glutamine. Moreover, control and mutant males showed similar PIs to L-alanine (appetitive effect) and to D-phenylalanine (aversive effect). To validate the role of the OBP19b gene in amino acid preference, we induced genomic rescue of the OBP19b gene in OBP19b-Gal4 > UAS-OBP19b flies. This was made possible by targeting the UAS-OBP19b transgene (containing the OBP19b coding sequence) with the OBP19b-Gal4 mutant transgene (containing the regulatory region of OBP19b gene). The resulting OBP19b-rescued males showed completely rescued PIs to both L-phenylalanine and L-glutamine (while their PI to L-alanine remained unchanged). We also compared the dose-dependent response of males presented to 2.5 and 25 mM L-phenylalanine or L-alanine (Fig. 4e and Supplementary Fig. 4). While control males, but not mutant males, showed an increased behavioral response to

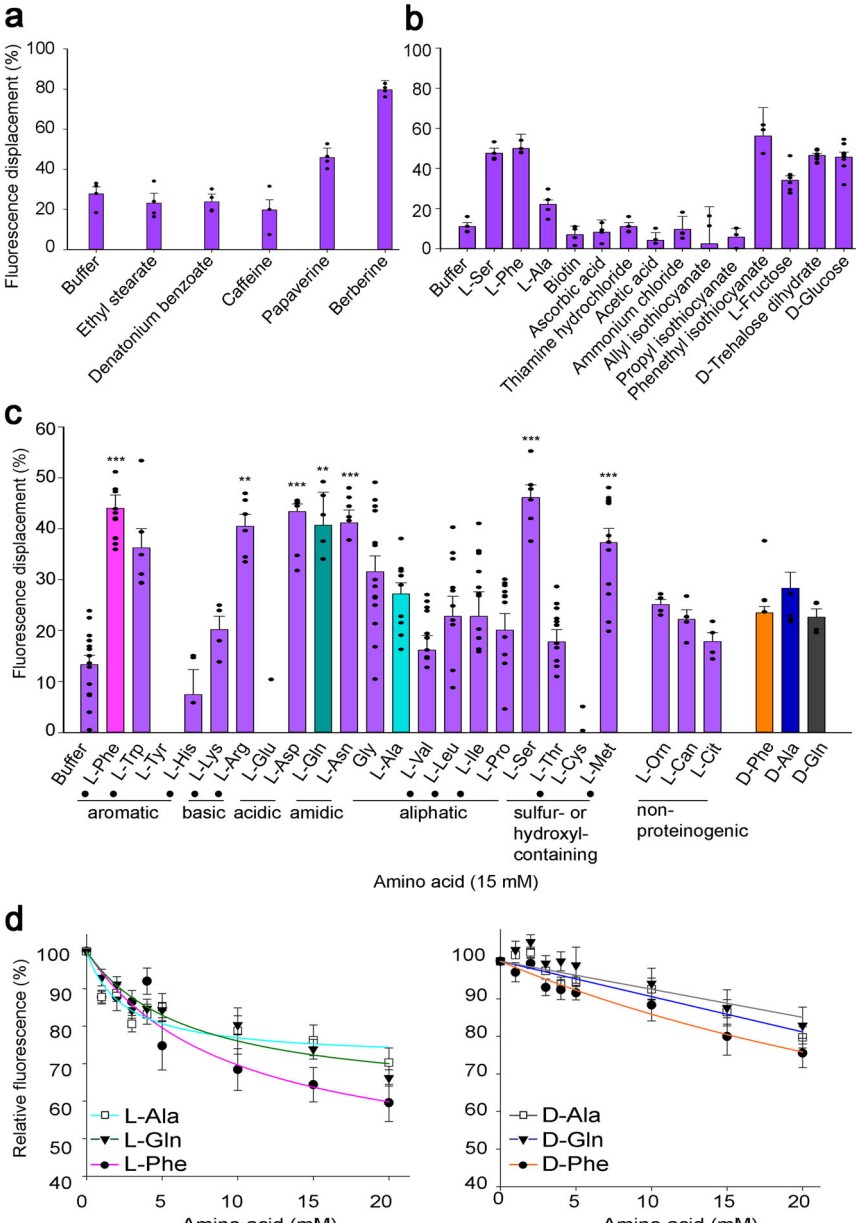

**Fig. 2 Binding properties of OBP19b with a competitive fluorescent assay.** The fluorescent displacement of the fluorescent probe NPN by various taste compounds was evaluated by adding aliquots of 50 mM ligand (dissolved in 50 mM sodium phosphate buffer at pH 7.5) to final concentration ranging from 0 to 20 mM. **a** Fatty acid esters and bitter compounds were tested at a final concentration of 40 μM. **b** Amino acids, vitamins, acids, amines, trigeminal (isothiocyanate), and sugars compounds were tested at a 15 mM final concentration. **c** Fluorescent displacement of 20 L-amino acids, including L-phenylalanine (L-Phe: magenta color), L-glutamine (L-Gln: green) and L-alanine (L-Ala: cyan); three nonproteinogenic amino acids L-ornithine (L-Orn), L-canavanine (L-Can), L-citrulline (L-Cit); and three D-amino acids (D-phenylalanine: orange, D-glutamine: blue, and D-alanine: gray). Dots indicate essential amino acids. Statistical differences were determined using one-way ANOVA followed by Dunnett's test (***$p < 0.001$; **$p < 0.01$). **d** Competitive-binding curves of L-Phe and L-Gln (showing the lowest $Kd_{app}$), L-Ala (negative control), and their three corresponding D-isomers. Data values represent the mean ± SEM. $N = 4–13$.

increasing L-phenylalanine concentrations, no such effect was noted to the increasing concentrations of L-alanine (Fig. 4e).

**OBP19b and electrophysiological response of taste sensilla.** Next, we determined the involvement of OBP19b in labellar taste sensilla in response to amino acid stimulation. We recorded the electrophysiological activity of the s6 sensillum (normal expression of OBP19b; Fig. 3d, e) in response to stimulation induced by specific amino acids (Fig. 5a). We compared action potential

spikes from control and OBP19b-null mutant flies. When stimulated with 10 mM L-phenylalanine, the recordings from the s6 sensilla of controls showed more spikes than recordings from the s6 sensilla of mutants. The difference in spike frequency between genotypes was reduced but remained noticeable with 3 mM L-phenylalanine (Fig. 5b, c). Given that sensilla from mutants showed no difference in the action potential values between the two L-phenylalanine concentrations, and that no difference in spike frequency was detected for the s6 sensilla of control and mutant flies in response to 10 mM D-phenylalanine (Fig. 5c), both

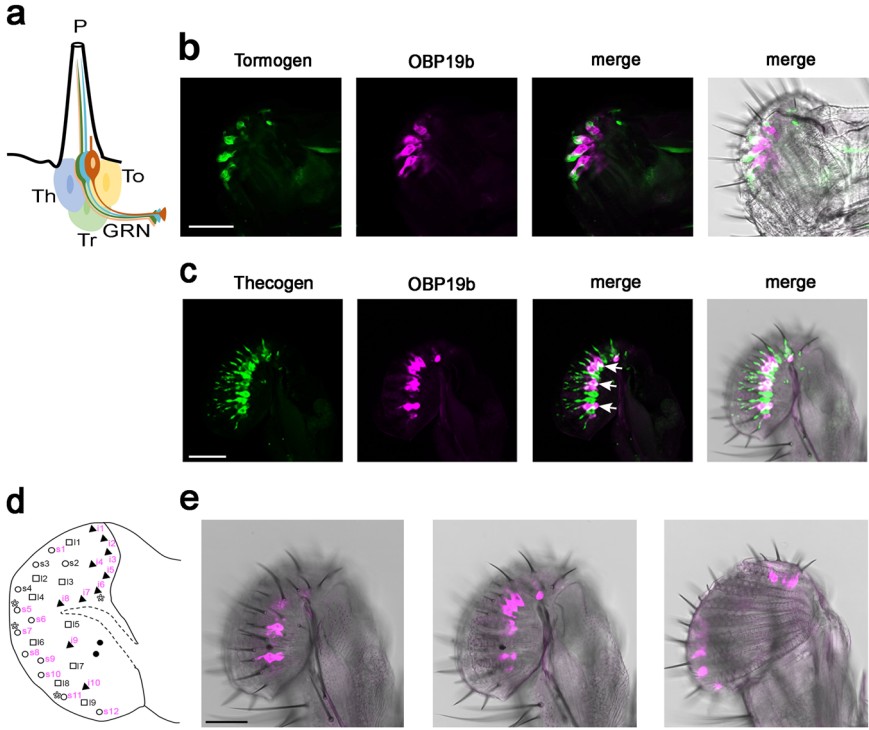

**Fig. 3** OBP19b expression in the proboscis. **a** Schematic representation of a Drosophila taste sensilla. Each sensillum ends with a terminal pore (P) and houses either two or four dendrites of gustatory receptors neurons (GRNs) and one mechanosensory neuron. Three accessory cells, thecogen (Th), tormogen (To), and trichogen (Tr), surround the GRNs. **b**, **c** OBP19b is expressed in the thecogen cells. The cellular distribution of OBP19b was examined based on the comparison and overlap (merge) of the spatial distribution of cell-type-specific markers (green) with OBP19b-Gal4 > UAS-mCherry transgenes (magenta). **b** Labeling of tormogen cells with ASE5-GFP. **c** Labeling of thecogen cells with *nompA*-GFP. Arrows point the most obvious overlapping between thecogen and OBP19b expressing cells. **d** Schematic representation of the repartition of labellum sensilla on the left hemi-proboscis. Taste sensilla are classified as a long (l; squares), intermediate (i; triangle), or short (s; circles) type depending on their length and relative position. The sensilla expressing OBP19b are labeled in magenta color. Asterisks indicate the sensilla containing a variable number (2–4) of taste neurons[57–59]. **e** Expression pattern of OBP19b in the thecogen cells of the proboscis sensilla. The three pictures represent three confocal layers taken from lateral to medial planes inside the left hemi-proboscis. $N = 50$, scale bars represent 50 μm.

these observations support the hypothesis that OBP19b confers stereospecificity for amino acid detection. We also compared the electrophysiological response of the s6 sensillum in control and mutant flies to four other amino acids (Supplementary Fig. 5). While a substantial difference was observed between genotypes toward 10 mM L-serine and L-glutamine (two amino acids shown to bind to OBP19b; Fig. 2c), no difference was detected to 10 mM L-tryptophan or L-alanine (which was shown not to bind OBP19b). The comparison of response to 1 mM KCl solutions and to amino acid solutions with or without tricholine citrate (TCC, used to inhibit water neuron activity[36]) indicates that TCC also inhibits amino acid responses (Supplementary Fig. 6). This is why we could not compare our s6 electrophysiological responses to L-phenylalanine with a previous study using TCC in the stimulating solution tested[37]. All the spikes were counted, including water spikes, revealing a big difference between electrophysiological responses of control and mutant flies towards L-phenylalanine, L-serine, and L-glutamine. Furthermore, the frequency of water spikes did not change between the two genotypes (Fig. 5c).

**Evolution and conservation of OBP19b.** Since most insects absolutely must detect essential amino acids, such as L-phenylalanine, in food sources, we reasoned that the mechanisms underlying such detection would have been conserved across evolution. By using a bioinformatics approach, we first found that the OBP19b protein sequence is not closely related to 14 other taste-associated OBP proteins of *D. melanogaster* (Supplementary Table 1). However, the comparison of the OBP19b protein sequence between species of the Drosophilidae family revealed a highly shared sequence identity[38] (~75%) with six conserved cysteine motifs (Table 2a and Supplementary Fig. 7a). Protein sequence conservation was also found among the heterologous sequences in ten non-Drosophilidae Diptera species (~40%; Table 2b and Supplementary Fig. 7b).

**Discussion**

Essential amino acids are indispensable in the diet of most animals. Amino acid deprivation in the *D. melanogaster* diet stops larval development and egg production[5,6]. Few mechanisms underlying amino acid detection and preference in insects have been discovered. The discoveries include the Drosophila Ir76b receptor, which modulates larval attraction to amino acids[12] and adult preference for amino acids[13]. OBPs were initially named according to their presence in olfactory tissues and their involvement in the processing of odorant molecules, although a few OBPs are expressed in gustatory tissues[19–23]. In gustation, OBP49a was shown to affect the detection of bitter molecules[28]. We observed a much higher expression of OBP19b in *D. melanogaster* gustatory organs compared to olfactory organs (31-fold). The restricted expression of OBP19b in gustatory appendages led to the suggestion that it had a role in taste detection. The low level of OBP19b expression in olfactory appendages is in agreement with the expression levels previously reported[39–42]. OBPs are proteins generally considered to transport hydrophobic

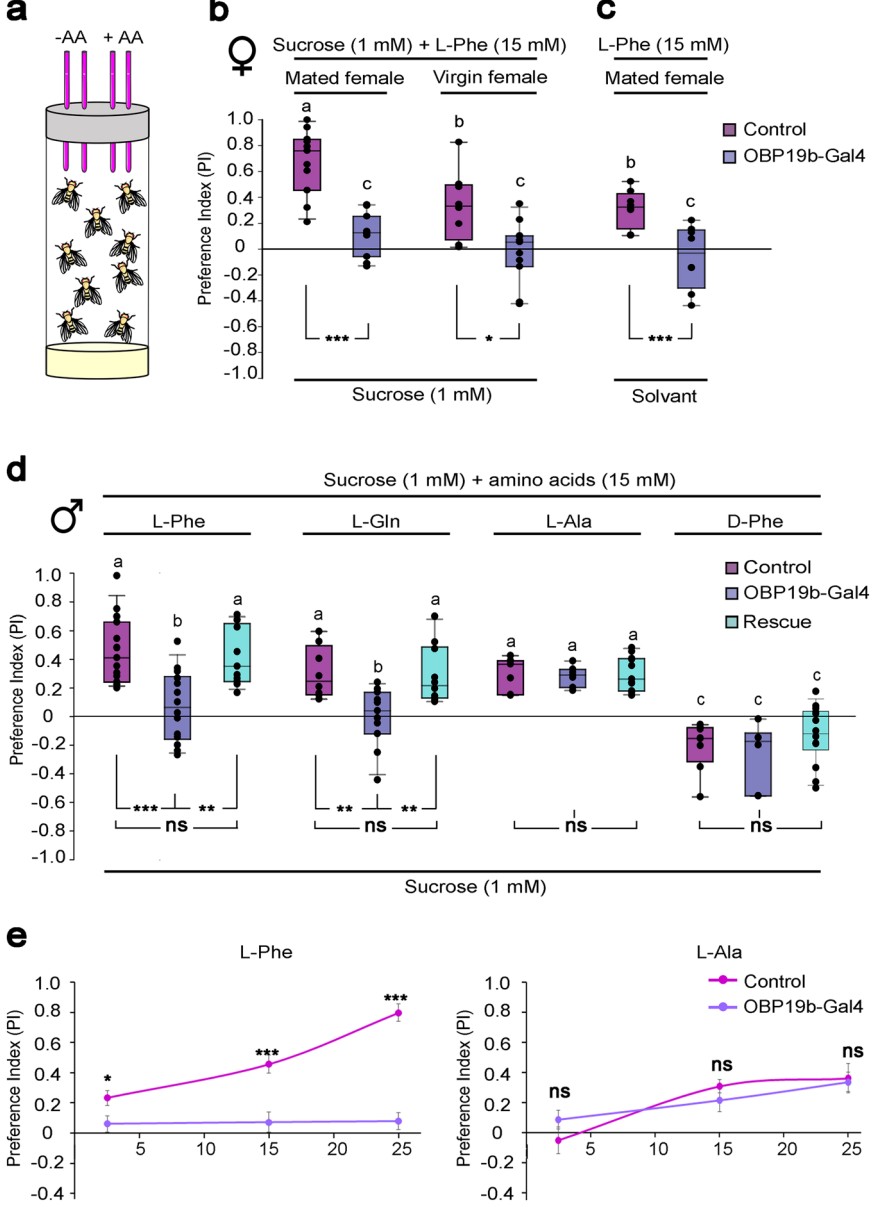

**Fig. 4 Amino acid taste preference. a** Schematic representation of the CAFE assay used to evaluate consumption preferences between a control solution without amino acid (-AA) and an amino acid-rich solution (+AA; 15 mM). **b** Preference for L-phenylalanine (L-Phe) by mated and virgin female flies of control and mutant OBP19b-Gal4 genotypes. Control and amino acid-rich solutions contained 1 mM sucrose. **c** Mated female preference for L-phenylalanine (L-Phe) solution in pure water (no sucrose). **d** Preference for L-phenylalanine (L-Phe), L-glutamine (L-Gln), L-alanine (L-Ala), and D-phenylalanine (D-Phe) by control, OBP19b-Gal4 null mutant and genetically rescued male flies (rescue: OBP19b-Gal4 > UAS-OBP19b). **e** Dose–response curves of control and OBP19b-Gal4 male flies in response to L-Phe and L-Ala. For each same-sex group (b/c or d), the letters indicate significant differences determined by the Kruskal–Wallis test and post hoc Wilcoxon test: ns = nonsignificant, *$p < 0.05$; ** $p < 0.01$; and ***$p < 0.001$. $N = 7–15$.

molecules. In this study, we found that OBP19b is involved in the detection of hydrophilic amino acids such as L-serine and L-glutamine, and hydrophobic amino acids such as L-phenylalanine. Structural studies of OBP19b-amino acid complexes would help to understand the molecular determinants of amino acid binding to OBP19b. The amino acid transport role of OBP19b corresponds to the classical function attributed to OBPs. In addition to its role in amino acid transport, and similarly to the LUSH OBP, OBP19b may also participate in the interaction with the protein receptor(s) involved in amino acid detection. Similarly unexpected roles of OBPs were also recently described for OBP28a and OBP59a, which are crucial for odorant buffering and humidity detection, respectively[29,39].

OBP19b, which can bind to several amino acids, showed the greatest affinity for L-phenylalanine and, to a lesser extent, for L-glutamine and L-serine. In contrast to olfactory OBPs, which bind odorants or pheromones with micromolar affinities, OBP19b binds amino acids with millimolar dissociation constants. This binding is stereospecific as indicated by the finding that OBP19b did not interact with D-amino acids and with several L-amino acids. Our behavioral and electrophysiological comparison of control and mutant flies indicates the specific role of OBP19b in the detection of some—but not all—amino acids. Moreover, the genetic excision of the OBP coding sequence (in mutant flies) together with its transgenic rescue supports our hypothesis that OBP19b is specifically involved in the detection of

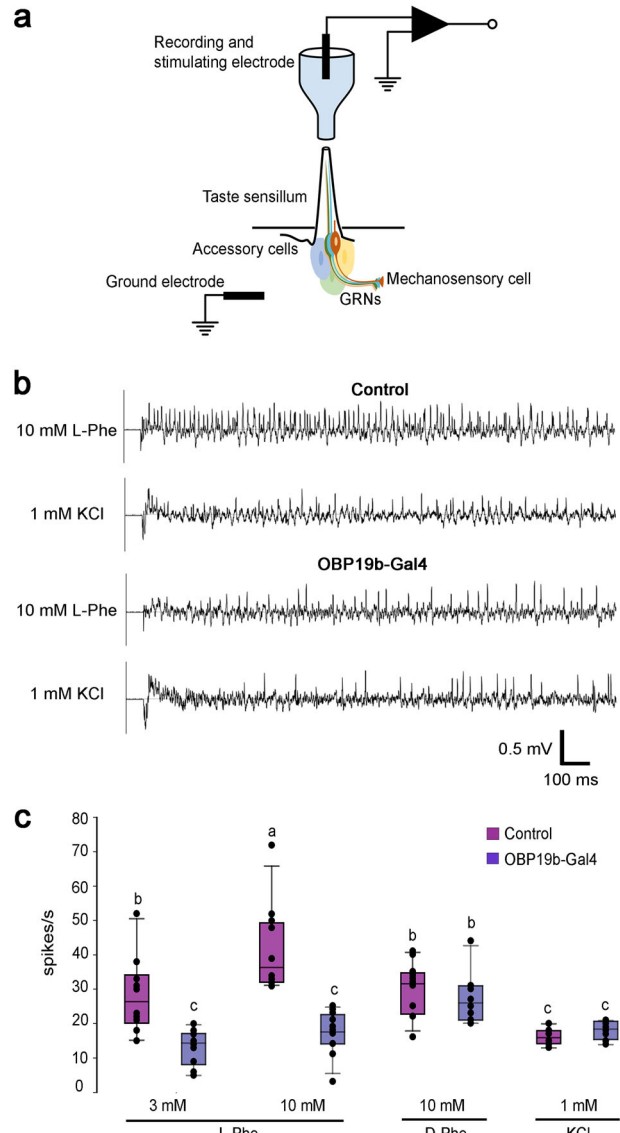

**Table 2 Evolutionary conservation of OBP19b.**

| (a) | | (b) | |
|---|---|---|---|
| **Species** | **% Identity** | **Species** | **% Identity** |
| *Drosophila simulans* | 93.9 | *Liriomyza sativae* | 45.2 |
| *Drosophila sechellia* | 92.4 | *Musca domestica* | 44.3 |
| *Drosophila yakuba* | 87.0 | *Calliphora stygia* | 43.4 |
| *Drosophila ananassae* | 84.7 | *Lucilia cuprina* | 43.0 |
| *Drosophila erecta* | 84.0 | *Bactrocera minax* | 40.0 |
| *Drosophila serrata* | 75.4 | *Ceratitis capitata* | 38.5 |
| *Drosophila pseudoobscura* | 73.6 | *Anastrepha obliqua* | 38.5 |
| *Drosophila persimilis* | 72.9 | *Anastrepha fraterculus* | 37.8 |
| *Drosophila willistoni* | 68.7 | *Bactrocera drosalis* | 37.7 |
| *Drosophila obscura* | 64.6 | *Delia antiqua* | 36.4 |
| *Drosophila virilis* | 58.8 | | |
| *Drosophila mojavensis* | 58.8 | | |
| *Drosophila grimshawi* | 56.9 | | |

(a) Percentage identity matrix showing protein sequence alignment of DmelOBP19b with the thirteen OBP19b proteins identified in Drosophilidae species by BLASTP (b) Percentage identity matrix showing protein sequence alignment of DmelOBP19b with ten OBPs in non-Drosophilidae Diptera

**Fig. 5 Electrophysiological responses of s6 sensillum in control and mutant flies. a** Schematic representation of the tip-recording method used to record the electrophysiological activity of a single sensillum on the Drosophila proboscis. The stimulus was contained in a glass microelectrode that capped the tip of the sensillum. Amino acids were dissolved in 1 mM KCl solution. Recordings were obtained from s6 sensilla in control CS and OBP19b-Gal4 mutant female flies. **b** Representative traces of tip-recording data obtained from control and OBP19b-Gal4 mutant flies after stimulation by 10 mM L-phenylalanine (L-Phe) and by 1 mM KCl. **c** Histograms representing the mean ± SEM for the number of spikes/s obtained in control and mutant flies in response to stimulation by 3 mM and 10 mM L-Phe, 10 mM D-phenylalanine (D-Phe) and by 1 mM KCl. L-Phe spikes might mask or interfere water response. Thus, the total number of spikes shown here is not a simple summation of spikes elicited by water and L-Phe. Letters indicate significant differences determined with ANOVA and Tukey's post hoc test (a/b or b/c: $p < 0.01$; a/c: $p < 0.001$); $N = 8$–12.

L-phenylalanine and L-glutamine but not in the detection of L-alanine or D-phenylalanine. However, we cannot rule out the possibility that other OBPs are involved in amino acid detection. Our data also support that amino acid sensing is mainly mediated by s sensilla[37]. Given that each s-type sensillum houses four GRNs (S, W, L1, and L2)[43], further studies are needed to identify the GRNs responding to amino acids.

OBP19b is expressed, and likely secreted, in the thecogen cells at the base of all i-type and most s-type proboscis sensilla. The fact that OBP19b was not found in any other gustatory appendage suggests that flies primarily use their proboscis to detect the presence and quality of amino acids in food. Moreover, in addition to the marked preference of all control flies for L-phenylalanine, the increased appetence shown by mated females, compared to virgin females and males, suggests that L-phenylalanine is also necessary for female reproductive function(s) such as egg production and egg laying[5,6]. The fact that OBP19b is also expressed—but at a much lower level—in olfactory appendages suggests that it could be also involved in odorant detection, but this remains to be elucidated.

The nutritional health of flies can affect their taste preference for amino acids. Amino acid deprivation increased ingestion of amino acids and the proboscis extension response to specific amino acids[7]. The intensity of the latter effect varied between the sexes and the amino acid presented: L-phenylalanine was one of the amino acids that induced the highest amplitude effect[7]. Moreover, amino acid deprivation from the diet increased the response of the fly taste pegs and sensillar GRNs to yeast[44]. The mated control females may have shown the greatest preference for 1 mM sucrose instead of water (Fig. 4b, c) because starvation increased the sugar sensitivity of Gr5a neurons[45]. However, the difference between control and mutant flies was not affected by the presence of 1 mM sucrose in the solution with the tested amino acids.

Although the internal level of amino acid affects the primary sensory neurons of the fly, it may also act in the brain. Indeed, Drosophila protein appetite can be modulated by, at least, two brain-related mechanisms: a small cluster of dopaminergic neurons enhances yeast intake in protein-deprived flies[46], and the protein-specific satiety hormone FIT inhibits protein-rich food intake[47]. It may be worth exploring the link between these central systems and the peripheral effect of OBP19b to better understand the modulation of protein appetite.

Finally, the low similarity of *D. melanogaster* OBP19b with other taste-expressed OBPs, taken together with its relatively high conservation with other Drosophilidae and Diptera species, indicates that this protein plays a crucial role in the detection of indispensable nutrients, such as L-phenylalanine, in other insects, at least those in the Diptera order.

**Table 3 Forward and reverse primers used for RT-qPCR.**

| *OBP* gene | Forward primer | Reverse primer |
|---|---|---|
| Obp19b | CTGCAACGAGGAGCTAAAGG | AGCATGCACATAGCGATCC |
| Obp19d | CACCGATGAGGATGTGGAG | TGTTCAGCTTACCGGATTCA |
| Obp28a | ACTGGTGCGAGCCTTTGA | CTCTCGGCTGACTCCATCA |
| Obp56e | TGCAGCTCTATCTTTGGCATC | GGCCTTGGCTCTCTGCTT |
| Obp57b | AGGCTCCCGAAGAACTTTGT | GGATGGCCAGCCTTAAATG |
| Obp83a | CTTCTGCTAAAAGCGAACGAG | CATCCGTGAAACAGCAAAAAT |
| Obp83b | ATTTGTGCTCCCAAAACTGG | CTCATGAATTTGCCCATCG |

## Methods

**Drosophila stocks.** All fly strains were raised on Drosophila standard medium under controlled conditions (24.5 ± 0.5 °C at 65 ± 5% relative humidity for a 12:12 h photoperiod during a subjective day between 8 a.m. and 8 p.m.). The OBP19b-Gal4 null mutant, *nompA*-GFP (BDSC_42694) and ASE5-GFP (BDSC_58449) lines were obtained from Bloomington Drosophila Stock Center. The UAS-OBP19b line (F003836) used to generate the rescued genotype was obtained from FlyORF. For the rescue, OBP19b-Gal4 homozygous females were crossed with UAS-OBP19b homozygous males to yield double heterozygous OBP19b-Gal4/+; UAS-OBP19b/+ F1 flies (OBP19b-Gal4>UAS-OBP19b). The wild-type strain Canton-Special (CS) was used as a control.

**RNA extraction and real-time quantitative reverse transcription PCR (RT-qPCR).** Total RNA was extracted using Isol RNA Lysis Reagent (5Prime) and treated with RNase-free DNase (Euromedex) to avoid genomic DNA contamination. Total RNA was reverse transcribed using the iScript complementary DNA (cDNA) Synthesis Kit (BioRad). The qPCR reactions were carried out on the MyIQ system (BioRad) using the IQ SYBR Green SuperMix (BioRad) and the primers described in Table 3. Each reaction was performed in triplicate. All results were normalized to actin and rp-49 mRNA levels and calculated using the DDCt method[48].

**Recombinant protein expression of OBP19b.** The cDNA encoding the mature OBP19b without its native signal peptide was amplified by PCR using cDNA of adult wild-type flies and the following primers: 5′ primer, 5′CCGCTCGAGAAA AGAGACGAGGAGGAGGGG, and 3′ primer, 5′ATAGTTTAGCGGCCGCTCA TTCCTTGATCTCGGGAATC. The synthetic cDNA of OBP19b was cloned into the *Xho*I and *Not*I sites of the pPIC9 plasmid, generating the construct pPIC9-OBP19b with the α-factor secretion signal fused to the mature OBP19b sequence without the Glu-Ala-Glu-Ala spacer peptide. The pPIC9-OBP19b plasmid was linearized with BglII and transferred into the GS115 strain of *Pichia pastoris* by electroporation, as described in the manual (version 3.0) of the Pichia Expression Kit (Invitrogen). The selection of the best secreting clone with the largest scale of protein production was completed as previously described[49]. OBP19b was secreted for only 4 h using a buffered minimal MeOH medium at pH 4.0 that was supplemented with 1% tryptone (Sigma-Aldrich) and 1% v/v of MeOH to promote induction. The yeast supernatant containing secreted OBP19b was clarified by centrifugation at 6000 rpm for 10 min at 4 °C and then was filtered (0.22 μm).

**Purification of recombinant OBP19b.** The supernatant containing OBP19b was first dialyzed at 4 °C against water for 4 h and then against 10 mM sodium acetate at pH 4.0 for 4 h and subsequently for another 4 h against 20 mM acetic acid at pH 4.0. Each dialysis step was repeated three times. The supernatant was then loaded onto a 5 mL SP-Sepharose 16 mm × 25 mm column (HiTrap SP HP, GE Healthcare). The column was washed with 25 mM acetic acid at pH 4.0, and the elution was performed using an increasing pH gradient, from 4.0 to 7.5, with 25 mM ammonium acetate (pH = 7.5). The selected fractions were dialyzed against 50 mM sodium phosphate and 150 mM NaCl at pH 7.5. OBP19b was concentrated and further purified by gel filtration using a Superdex 75 10/300 GL column. The selected fractions were dialyzed extensively against 50 mM sodium phosphate at pH 7.5 and stored at −20 °C. SDS-PAGE analysis was performed according to the method of Schägger and von Jagow[50].

**Biophysical analysis.** OBP19b was analyzed by MALDI-ToF mass spectrometry with a Voyager DE-Pro (Sciex) spectrometer using a positive linear mode. Circular dichroism spectra were recorded using JASCO spectrometer J-815 equipped with a Peltier temperature control system (JASCO MPTC-490S) and a 0.1 mm thick quartz cell. Measurements were taken at a protein concentration of 5.0 mg/mL in 50 mM sodium phosphate at pH 7.5. Ten measurements were recorded for each spectrum over a range of 180 to 260 nm at 20 °C with a data pitch of 0.5 nm and a scanning speed of 100 nm/min. The contribution of the buffer was corrected for and converted into molar ellipticity.

**Fluorescence-based-binding assay.** Competitive ligand-binding experiments were performed using *N*-phenyl-1-naphthylamine (NPN) as a fluorescent probe[51]. Emission fluorescence spectra were recorded using a Cary Eclipse spectro-fluorometer (Agilent Technologies) with a 1 cm light path, quartz cuvette and 5 nm slits for both excitation and emission. To determine the binding affinity of OBP19b for NPN, a 2 μM protein sample in 50 mM sodium phosphate buffer at pH 7.5 was titrated with aliquots of 10 mM NPN in 10% methanol to a final concentration of 0.5–10 μM. The excitation wavelength of NPN was 337 nm, and the fluorescence emission was recorded between 380 and 450 nm. The value of the dissociation constant of the complex OBP19b/NPN ($K_{NPN}$) was calculated using SigmaPlot software via nonlinear regression of a unique binding site.

To measure the affinity of OBP19b to ligands, a competitive screening assay was performed at 23 °C using a 96-well microplate format and a Victor3 V microplate reader (PerkinElmer Life Sciences) with 25 and 5 nm slits for excitation and emission, respectively. The excitation wavelength was 340 nm, and the fluorescence emission intensity was recorded at 415 nm. First, a wide screening array involving various ligand compounds was performed. Each well contained 200 μL of 2 μM OBP19b with 4 μM NPN dissolved in 50 mM sodium phosphate buffer at pH 7.5. Fatty acids and bitter compounds were tested in the μM range, whereas sugars, amino acids, vitamins, acids, amines, and isothiocyanate compounds were tested in the mM range. All tested compounds were dissolved in 10% methanol except amino acids, which were dissolved in 50 mM sodium phosphate buffer at pH 7.5. Based on the results of the first screen, the 20 L-amino acids, certain D-amino acids and three nonproteinogenic amino acids were tested by adding 100 μL of a 30 mM ligand stock solution to the microplate wells to obtain a final concentration of 15 mM, and then the solution was mixed. Significant differences using one-way analysis of variance (ANOVA) followed by Dunnett's test were observed compared to the buffer (***$p < 0.001$; **$p < 0.01$). The competitive-binding curves were identified by titration of OBP19b/NPN complex with aliquots of 50 mM ligand to final concentration ranging from 0 to 20 mM. The dissociation constants ($Kd_{app}$) of each compound were calculated according to the corresponding $IC_{50}$ values (the concentration of ligands required to halve the initial fluorescence value of NPN) using the following equation: $Kd_{app} = [IC_{50}]/(1 + [1 − NPN]/K_{NPN})$, where [NPN] is the free concentration of NPN. The reported $Kd_{app}$ values are the average of three measurements performed on three independently experiments.

**CAFE assay.** Flies were collected from 0 to 8 h after hatching and kept in fresh food vials. When flies were 5- to 7-days-old, ten flies were starved for 18 h in vials with a wet piece of cotton placed at the bottom. The amino acid consumption by male, virgin and mated female flies was evaluated separately using a capillary feeder[52]. Four 5-μl minicaps (Hirschmann Laborgerate, GmbH & Co.) filled with the test solutions were inserted through the caps of the vial. Flies were presented two choices: a minicap filled with 1 mM sucrose or solvent that served as the control solution and one minicap filled with 15 mM amino acid and 1 mM sucrose or one minicap filled with only 15 mM amino acid. Test solutions were diluted in 50 mM sodium phosphate buffer at pH 7.5 and colored with the red dye sulfor-hodamine B (3520-42-1; Sigma-Aldrich) to easily visualize the surface boundary of the test solution. Mineral oil was overlaid on the minicaps to prevent evaporation. The feeding test lasted 6 h under white light at room temperature (25 °C) and 50–60% humidity. Simultaneously, for each series of tests, we included a similar CAFE device without flies to determine the evaporation level, which was subsequently subtracted from the experimental levels measured. All experiments were performed between 8 a.m. and 2 pm. Images of capillaries were captured before and after the test, and the amount of food intake was measured using ImageJ software. Amino acid consumption was evaluated with a preference index (PI) according to the following equation: $PI = (V_{aa} − V_c)/(V_{aa} + V_c)$, where $V_{aa}$ and $V_c$ represent the volume of the amino acid-rich and the control solution consumed, respectively. PI values ranging between +1.0 and −1.0 indicate a preference for the amino acid-rich solution or for the control solution, respectively, while PI=0 indicates the absence of preference (or indifference). Statistical analysis was performed with the Kruskal–Wallis test to compare same-sex treatments followed by a post hoc Wilcoxon test to assess two-by-two differences between genotypes and tested amino acids.

**Electrophysiology recording**. Recordings from a single sensillum of 2–3-day-old flies were made using the tip-recording method[53,54]. In brief, neuronal responses were obtained from mated female flies after inserting a glass capillary filled with adult hemolymph-like saline through the dorsal thorax and neck to the inside part of the proboscis. The adult hemolymph-like saline contained 108 mM NaCl, 5 mM KCl, 2 mM $CaCl_2$, 8.2 mM $MgCl_2$, 4 mM $NaHCO_3$, 1 mM $NaH_2PO_4$, 5 mM trehalose, 10 mM sucrose, and 5 mM HEPES (pH 7.5 and 265 mOsm)[55]. A single sensillum was stimulated for 2 s by a second glass pipette (recording electrode) containing the experimental taste-stimulating solution containing 1 mM KCl used as the electrolyte. Recording was started upon contact, with at least 1 min between presentations. Electrical signals were sampled at 10 kHz on a computer and analyzed using Clampfit software to detect and sort the action potential values (spikes). As the tricholine citrate used to inhibit water neuron activity in our hands also inhibited amino acid responses (Supplementary Fig. 6), we could not compare the s6 electrophysiological responses to L-phenylalanine we obtained with those recently reported[37]. The sensilla responses were quantified by counting the number of spikes over a 1 s period starting 100 ms after contact. As a positive control, we checked the response to low- and/or high-salt stimulus (50 and 400 mM NaCl, respectively) at both the beginning and at the end of each recording session for each sensillum[56]. All solutions were kept at 4 °C and were used within 1 week. Statistical differences in the number of spikes/s were determined with ANOVA and Tukey's post hoc test ($p < 0.001$). Values are given as the mean ± SEM.

**Immunohistochemistry**. Labella from 5- to 7-day-old male flies were dissected and drilled using a pin in a buffer solution (1x PBS and 0.2% Triton X-100). Tissues were incubated in fixation buffer (PFA 4%, 1X PBS, and 1% Triton X-100) for 45 min, agitated at room temperature (RT) and washed (1x PBS and 1% Triton X-100) for 15 min. In toto, labella were transferred through blocking buffer (0.5% blocking reagent, 0.15 M NaCl, and 0.1 M Tris HCl at pH 7.5) for 45 min under agitation at RT. The whole-mount fly labella was first incubated in goat anti-GFP (1:500, Rockland) and rabbit anti-mCherry (1:1000, Rockland) primary antibodies for 48 hours at 4 °C and washed 5 times for 15 min with 1x PBS and 0.2% Triton. Thereafter, tissues were incubated overnight in anti-goat-Alexa 488 (1:400 antibody, Molecular probes) and anti-rabbit-Alexa 594 (1:400 antibody, Molecular probes) secondary antibodies in the dark at 4 °C. Tissues were then washed with 1x PBS and 0.2% Triton five times for 15 min and were mounted with Vectashield H-1200 in DAPI medium. The labeling was visualized with a Leica sp8 confocal microscope and analyzed using ImageJ software (Software, NIH, Bethesda, MD, USA).

**Protein sequence analysis**. To identify putative non-annotated OBP19b members, we first conducted a BLASTP search against the genome sequence of Diptera species. The OBP19b protein sequences of Drosophilidae that were identified were retained for further analysis. In addition, based on the BLASTP analysis, the OBP protein sequences identified in Diptera that shared >35% identity with the protein sequence of DmelOBP19b were also selected. The signal sequence-bearing N-terminus was identified and removed from each OBP sequence. Then, the protein sequence of DmelOBP19b was aligned to the selected sequences of other species, and the sequence identity matrix was calculated using CLUSTAL Omega alignment (https://www.ebi.ac.uk/Tools/msa/clustalo/).

**Statistics and reproducibility**. The competitive-binding assay data were compared to the buffer using one-way ANOVA followed by Dunnett's test (***$p < 0.001$; **$p < 0.01$), the experiments were reproducible, $N = 4$–13. The evaluation of statistical significance of differences of the CAFE assay data was performed with Kruskal–Wallis test to compare same-sex treatments and post hoc Wilcoxon test to assess two-by-two differences between genotypes and tested amino acids. The experiments were reproducible and for each Drosophila strain the experiments were repeated 7 to 15 times (each tube containing ten flies). The electrophysiological data were analyzed by comparing the number of spikes/s between genotypes and tested amino acids using ANOVA and Tukey's post hoc test ($p < 0.001$). The experiments were reproducible and for each Drosophila strain the experiments were repeated eight to 12 times.

**Reporting summary**. Further information on research design is available in the Nature Research Reporting Summary linked to this article.

## Data availability

All relevant data supporting the findings of this study are available from the corresponding author on request. Source data underlying plots are provided in Supplementary Data 1.

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

## Acknowledgements

We thank Bloomington Drosophila Stock Center, FlyORF Zurich ORFeome Project, Seok Jun Moon and Craig Montell for *D. melanogaster* stocks, Claude Everaerts for help with the statistics and DIMACell platform for microscopy studies (University of Burgundy, Dijon, France). Mass spectrometry experiments were performed by the Plateforme SFR BioSciences (UMS 3444), Protein Science Facility (PSF) of the Institut de Biologie et Chimie des Protéines (IBPC CNRS/Université de Lyon, 7 Passage du Vercors 69367 LYON Cedex 07).

## Author contributions

K.R., T.T., J.F.F., and L.B. designed the research. K.R., S.F., N.P., T.D., and I.C. performed the research. K.R., S.F., I.C., and F.N. analyzed the data. The manuscript was written by K.R., T.T., F.N., J.F.F., and L.B. All authors read and approved the final manuscript.

## Competing interests

The authors declare no competing interests.
