## [Peer Review File · Communications Biology]

Reviewers' comments:

Reviewer #1 (Remarks to the Author):

Rihani et al. present a well-rounded manuscript which begins with identifying the protein Obp19b as a candidate for gustatory function in taste sensilla of the proboscis. To investigate the structure and function of Obp19b, Rihani et al. express this protein in yeast using a robust method, and employ the well-documented fluorescence competitive binding assay to determine potential ligands. This panel of ligands is then further refined through a series of electrophysiological and behavioral methods to identify the physiologically relevant binding partner(s) of Obp19b.

Rihani et al. employ various tools in their investigation, creating a thorough story that is certainly of use to others in the field. Their conclusion that Obp19b is necessary for sensation of a very narrow range of tastants is novel, and challenges the prevailing theory that all Obps function to carry odorants to receptors in the olfactory system. Not only does this manuscript identify a new molecular target for insect control, but it provides insight into the evolution of the taste system in insects.

While this paper is well thought out and addresses the central question using complementary techniques, there are several issues that need to be addressed before publication:

1. In Supplementary Figure 1, the range of [NPN] used should extend beyond what is displayed here, since the plateau for fluorescence intensity saturation has not yet been reached. This could impact the dissociation constant you calculated here, as evidenced by the high standard error. In turn, your conclusions about binding partners could also be misguided.
2. In Supplementary Figure 2, L-Gln and L-Ser show a comparable K_d value to L-Phe. It would be prudent for Rihani et al. to reinforce their findings with L-Phe by consistently exploring L-Gln and L-Ser response in the electrophysiological and behavioral assays.
3. In Table 1, were these the only 7 Obps detected? If not, it would be useful to the public for all the transcriptomic data to be presented in Supplementary Data.
4. In Figure 2 a/b, Rihani et al. explained that the ligand concentration used in these binding assays was determined by previous electrophysiological work. However, the assay described here is *in vitro* with a purified amount of Obp19b and ligand, neither of which may accurately represent physiological concentrations. Perhaps some of these identified ligands are false positives because the concentration of ligand is simply too high [and not physiologically relevant]? As such, I recommend that the authors test different concentrations of both protein and ligand here, to present a refined picture of Obp19b function.
5. In Figure 3, it is not clear that Obp19b is expressed in thecogen cells, given the image presented here. Can you say why the two markers do not overlap more? The authors should make a comment about whether they think Obp19b is expressed in trichogen cells, for which they do not have a marker.
6. In Figure 4, a dose-response curve is absolutely necessary for this behavioral assay, especially since some compounds become aversive or attractive at differing concentrations. Perhaps the lack of difference seen in L-Ala is because of the dosage which was tested - you might see a difference at a lower concentration? Also, why is D-Phe not tested in the rescue? On that note, what is the rationale for splitting genders and only testing L-Phe with females while males are tested against more AAs?
7. Lines 213-214, what is the evidence for Obp-IR interaction? There was no text in the results which backed up this claim.
8. Line 216, it might help your argument to add in the reference to Obp28a's unexpected

buffering role (Larter et al.).

9. Line 217, could this be an artifact of in vitro expression?

10. Line 227, "GRN" instead of "GNR"

11. Line 220, "OBP" instead of "OPB"

12. Line 235, other Obps could also detect AAs, but you haven't ruled that out with the experiments presented here.

13. Line 250, what about Obp19b in the antenna? What could its role be in other organs?

-- Dr. Jennifer S. Sun

Reviewer #2 (Remarks to the Author):

In this study, *Drosophila* OBP19b was shown to bind to some amino acids, and a mutant for Obp19b gene was defective to show preference to these amino acids in feeding choice assay. Furthermore, and most surprisingly, these amino acids were shown to activate bitter-sensing neurons and OBP19b was necessary for it. These results seem inconsistent within themselves, but (although it may sound paradoxically) that makes this study so original and important that the finding should be shared by others immediately, which may stimulate further studies on this unexpected fact.

I think the experiments were performed more than satisfactorily, considering the wide range of approach from in vitro to in vivo assays, which is at the highest standard in the recent OBP studies. Nevertheless, the presentation of their results and conclusions are not perfect, in my opinion, as listed below.

1. The results shown in Fig.2b, Fig.2c, and Fig.2d are not very consistent to each other. In particular, why L-serine displaced NPN so little in Fig.2b while it did so much in Fig.2c? Any technical reasons?

2. Related to the above comment, number of replicates was not shown in the all experiments except Table 1. What does the error bar indicate in each figure? It concerns how your results are reproducible.

3. In the binding assay shown in Fig.2b and d, what was the volume (or concentration) of the stock solutions of ligands?

4. Obp19b[1] = Obp19b-Gal4, but it is not obvious to the most readers (who may think Obp19[1] is a classical EMS allele). Should be explicitly explained that Obp19b[1] is a gene-targeting mutant in which Obp19b ORF was replaced with Gal4. 'Obp19b[1]>UAS-GFP' makes no sense.

5. In Fig.3c, I don't see the expression is not so much overlapping. Both GFP and mCherry didn't have any sub-localization signal? Then why it looks like this? It may be more convincing to say OBP19b is expressed in trichogen.

6. nompA-Gal4? It must be nompA-GFP otherwise the experiment shown in Fig.3c was impossible, but described as 'nompA-Gal4>UAS-GFP' in Materials & Methods and Fig.3 legend.

7. For the rescue experiment shown in Fig.4d, it is better to show the exact genotype of the rescued fly instead of Obp19b[1]>UAS-Obp19b. Unlike an usual crossing scheme for Gal4/UAS expression system, Obp19b-Gal4 needs to be homozygous in this experiment (UAS-Obp19b can be heterozygous). An explicit description is preferable as well as that for Obp19b[1] =

Obp19b-Gal4.

8. Results from electrophysiological analysis are really confusing. But I dare to say this is great because it does not look 'beautiful'. Nevertheless, the finding is so surprising from our current understanding in taste perception that some more discussion needs to be presented. Do you want to deny the 'labeled-line' model for taste perception with this finding? I suppose not. Then, readers may reasonably expect more thorough description about what kind of mechanisms can explain these self-inconsistent results, which should be supported by known facts. Or, if there are any technical limitations in the experimental design, they should be presented honestly with discussions about how they could influence the results and interpretation of them. I think it does not compromise any of this study's value.

Reviewer #3 (Remarks to the Author):

Rihani and co-workers present an original work on the role of OBP19b on AA detection. This subject is very interesting as OBPs form an heterogeneous family of proteins that has been originally described as binding odorant but since shown has having diverse roles. It has been extensively developed recently partly through the work of Carlson's group, notably from their study of 2016 in Elife (Larter & Al.)

While the study is globally well written and well conducted, some data could be enhanced and expanded to better support the author's conclusions. It holds true from the sensilla localization of OBP19b to the behavioural and electrophysiological characterization of the mutant impact on taste modality. Below are major and minor comments.

Major:

1) The author's results should be compared with the recent study from Park and collaborators in 2017 focused on AA detection in *Drosophila*. Indeed, in the Park study, Phe elicits mostly response from S5, S7 and S9 and poor one from S6. The rationale of focusing on s6 sensilla should thus be discussed.

2) Similarly, in the Park study, Trp elicits strong response in S6, and Rihani and collaborators observe in the current study that Trp seems to be effectively bounded by OBP19b (even if it must be at the threshold of significance (very similar results to L-Phe in Fig.2). The present work would thus gain in clarity by exploring the detection of additional AA such as L-Trp or L-Ala. L-Trp detection should be very similarly affected to L-Phe in OBP19b mutant (at both electrophysiological and behaviour level). Additionally, exploring the electrophysiological response to L-Ala would be interesting as the behaviour detection of L-Ala which binds to the OBP according to Fig 2.D is not affected by in the OBP19b mutant. If the authors cannot provide such data, it should at least be discussed and temper their conclusion about the specificity of OBP19b toward L-Phe detection.

3) In the OBP19b binding assay, the rationale of using sugars at μM concentration but AAs in the mM range should be explained are both are considered as nutrients.

4) As it is, the authors provide insufficient data to clearly establish which sensilla express OBP19b. Is it stable on different flies? How many did they examine?

5) The electrophysiological dose responses to L-Phe contrast a lot with previous study (Park 2017). The basal firing is very high at 3 mM (similarly as recordings with the ineffective behaviourally D-Phe) and increase only slightly at 10 mM. Recordings with lower concentration showing the basal rate of firing would help to demonstrate the specificity of the response. Could this high basal firing rate be the response of the water cell? The authors explain in the M&M section that the water cell in s6 may only be active in mutant, which indicates important difference in basal function of taste sensilla and make the comparison between WT and mutant difficult. It is also puzzling that the KO of OBB19b seems to reduce this basal firing rate with L-Phe recordings but not with D-Phe. The authors should therefor temper their conclusions as

in the discussion; they state "Our behavioral and electrophysiological comparison of control and mutant flies validates the specific role of OBP19b in the detection of L- but not D-phenylalanine. Moreover, the genetic excision of the OBP coding sequence (in mutant flies) and the transgenic rescue of it validated the speculation that OBP19b is specifically involved in the detection of L-phenylalanine and L-glutamine but not the detection of L-alanine or D-phenylalanine."

6) Line 231: " Taken together, these results indicate that bitter-sensing GRNs can send information leading to either appetitive or aversive behavior." They authors should be much more cautious in their conclusion as it seems very counter intuitive to imagine that if L-Phe is only detected by bitter sensing neurons then their activation can lead to opposite behaviour when compared to activation with bitter compounds. In order to present this statement; they should genetically eliminate bitter sensing neurons and observe that both L-Phe detection in s6 sensilla and behaviourally attraction is abolished.

7) The mixture of caffeine and L-Phe in tip recording is consistent with the fact that L-Phe and caffeine are probably detected by the same cell but is not a real demonstration. Indeed, they authors present only one example of recording and no real spike analysis such as PSTH or ISI. It is however consistent with the Park study of 2017 showing that sensilla housing a bitter responding cell genetically ablated could not detect any more Trp. The authors should be more cautious in their assessment.

8) Could the authors provide a reference showing that an approximately 75% percentage of identity indicates an highly conserved sequence? 75% seems rather average considering this comparison is limited to Drosophila species

Minor points:

1) In the introduction, the recent work on OBP particularly of Carlson team having both published mainly on AA detection and OBP roles is not mentioned (Larker 2016, Park 2017 and Sun 2018). It would be interesting to expand the introduction on the role that OBP may play outside olfaction such as in humidity detection and not only mention part of them in the discussion.

2) The rational of using either virgin or mated females in the CAFE assay should be described before the discussion to improve the readability of the manuscript.

3) line 26 " ionotropic receptor-expressing gustatory receptor neurons". Could you improve the readability of this part?

4) On figure 3e: The sensilla expressing OBP19b should be annotated to link them better with the confocal images. Maybe using colour?

5) Line 47: Precise the model "The general control nonderepressible 2 pathway detects the absence of AAs, whereas the Target of Rapamycin kinase pathway monitors particular AAs"

6) Line 54 : The absence of umami taste is puzzling when speaking of AA detection in mammals. "Many AAs attract rodents and elicit a savory or sweet taste response in humans"

7) Line 211: "In this study, we found that OBP19b is involved in the detection of hydrophilic molecules such as AAs". It seems a bit strange to qualify AA such as L-Phe of hydrophilic molecule. In fact, the authors should discuss how OBP19d could bind similarly Glu and Phe which are polar and hydrophobic respectively.

8) Some part of the discussion could be shortened as they are quite speculative, notably the 3 paragraphs line 234, 241, 250. It would rather be interesting to expand the discussion on how their study fit in the model of OBP function. Is it to improve AA solubility? Similarly, it could be

interesting to compare their results according to recent works on the roles and distribution of OBPs in *Drosophila* organs. For example, in the study from Larker 2016, the authors observed that OBP19b is not expressed in OSN similarly as the present study.

Answers to the reviewer's comments on the manuscript entitled: “A conserved odorant binding protein is required for essential amino acid detection in *Drosophila*” by Karen Rihani, Stéphane Fraichard, Isabelle Chauvel, Nicolas Poirier, Thomas Delompré, Fabrice Neiers, Teiichi Tanimura, Jean-François Ferveur and Loïc Briand.

Manuscript ID: COMMSBIO-19-0657-T

We wish to thank the reviewers and the editor for their insightful comments, which significantly helped us to complete the experiments to improve our manuscript.

Referee expertise:

Referee #1: OBP

Referee #2: OBP and behavior in *Drosophila*

Referee #3: Olfaction

Reviewers' comments:

Reviewer #1 (Remarks to the Author):

Rihani et al. present a well-rounded manuscript which begins with identifying the protein Obp19b as a candidate for gustatory function in taste sensilla of the proboscis. To investigate the structure and function of Obp19b, Rihani et al. express this protein in yeast using a robust method, and employ the well-documented fluorescence competitive binding assay to determine potential ligands. This panel of ligands is then further refined through a series of electrophysiological and behavioral methods to identify the physiologically relevant binding partner(s) of Obp19b.

Rihani et al. employ various tools in their investigation, creating a thorough story that is certainly of use to others in the field. Their conclusion that Obp19b is necessary for sensation of a very narrow range of tastants is novel, and challenges the prevailing theory that all Obps function to carry odorants to receptors in the olfactory system. Not only does this manuscript identify a new molecular target for insect control, but it provides insight into the evolution of the taste system in insects.

While this paper is well thought out and addresses the central question using complementary techniques, there are several issues that need to be addressed before publication:

1. In Supplementary Figure 1, the range of [NPN] used should extend beyond what is displayed here, since the plateau for fluorescence intensity saturation has not yet been reached. This could impact the dissociation constant you calculated here, as evidenced by the high standard error. In turn, your conclusions about binding partners could also be misguided.

Our answer: The authors agree with the referee's comment that the range of NPN needed to be extended. To clarify this point, we performed a titration curve of OBP19b with NPN at higher NPN concentrations to reach the fluorescence intensity saturation. With this extended curve, we calculated a new dissociation constant value ($4.1 \pm 0.9 \mu\text{M}$) and also deduced the dissociation constants for ligands. We changed the manuscript (line 112) and the **Supplementary Figures 1 and 2**.

2. In Supplementary Figure 2, L-Gln and L-Ser show a comparable K_d value to L-Phe. It would be prudent for Rihani et al. to reinforce their findings with L-Phe by consistently exploring L-Gln and L-Ser response in the electrophysiological and behavioral assays.

Our answer: We have followed the referee's remark and reinforced our findings with additional experiments. More precisely, we recorded the electrophysiological

responses to L-Gln and L-Ser. The OBP19b deletion revealed an effect similar to that observed with L-Phe. These novel data are now shown in a new **Supplementary Figure 5**. A paragraph was added in the Results section (lines 194-198): “ We also compared the electrophysiological response of the s6 sensillum in control and mutant flies to four other AAs (**Supplementary Figure 5a, b**). While a significant difference was observed between genotypes toward 10 mM L-serine and L-glutamine (two AAs shown to significantly bind to OBP19b; **Figure 2c**), no difference was detected to 10 mM L-tryptophan or L-alanine (which was shown not to bind OBP19b)”.

3. In Table 1, were these the only 7 Obps detected? If not, it would be useful to the public for all the transcriptomic data to be presented in Supplementary Data.

Our answer: We did not perform transcriptomic analysis. The data reported in Table 1 have been obtained using RT-qPCR. Given that the legend of Table 1 mentioned in the previous version “transcriptomic comparison”, to avoid confusion we changed the legend of Table 1 as follows (lines 619-620): “Expression levels of OBP transcripts measured by RT-qPCR in head olfactory and taste appendages of adult male *Drosophila melanogaster*.”

4. In Figure 2 a/b, Rihani et al. explained that the ligand concentration used in these binding assays was determined by previous electrophysiological work. However, the assay described here is in vitro with a purified amount of Obp19b and ligand, neither of which may accurately represent physiological concentrations. Perhaps some of these identified ligands are false positives because the concentration of ligand is simply too high [and not physiologically relevant]? As such, I recommend that the authors test different concentrations of both protein and ligand here, to present a refined picture of Obp19b function.

Our answer: We want to clarify that, by definition, the dissociation constant (K_d value) is independent from protein (OBP) concentration. In consequence, it is not necessary to test different concentrations of proteins (OBPs). Concerning the hypothesis of false positive interaction raised by the referee, we believe that this hypothesis is unlikely since the binding of amino acids onto OBP19b is stereospecific (for instance, L-phenylalanine induced significant fluorescence reduction of OBP19b-NPN complex whereas D-phenylalanine has no significant impact on NPN fluorescence).

5. In Figure 3, it is not clear that Obp19b is expressed in thecogen cells, given the image presented here. Can you say why the two markers do not overlap more? The authors should make a comment about whether they think Obp19b is expressed in trichogen cells, for which they do not have a marker.

Our answer: Following the referee’s remark, we performed new immunohistochemistry experiments. The pictures have been substituted in **Figure 3c** and **Figure 3e**.

6. In Figure 4, a dose-response curve is absolutely necessary for this behavioral assay, especially since some compounds become aversive or attractive at differing concentrations. Perhaps the lack of difference seen in L-Ala is because of the dosage which was tested - you might see a difference at a lower concentration? Also, why is D-Phe not tested in the rescue? On that note, what is the rationale for splitting genders and only testing L-Phe with females while males are tested against more AAs?

Our answer: As suggested by the referee, we performed additional behavioral assays to evaluate the dose-dependent response of control and mutant flies to 2.5, 15 and 25 mM L-Phe and L-Ala. Our new data indicate that there is a dose-dependent response for L-Phe in control flies but not in OBP19b-Gal4 null mutant.

In contrast, no dose-response was found to L-Ala either in the control or in OBP19b-Gal4 null mutant flies at the three tested concentrations.

As requested by the referee, we also tested the preference of D-Phe with rescued flies. The **Figure 4d** was modified accordingly and we added the **Figure 4e**. A **Supplementary Figure 4** was also added to show the preference indexes at 2.5 and 25 mM L-Phe and L-Ala. The following paragraph has been added in the Result section (lines 176-180): “We also compared the dose-dependent response of males presented to 2.5 and 25 mM L-phenylalanine or L-alanine (**Figure 4e, Supplementary Figure 4**). While control males, but not mutant males, showed an increased behavioural response to increasing L-phenylalanine concentrations, no such effect was noted to the increasing concentrations of L-alanine (**Figure 4e**).”

Previous studies have shown sexually dimorphic feeding preference for AAs (Ganguly et al., 2017). To clarify this point we added this sentence in the Results section (lines 153-154): “Previous studies have shown sexually dimorphic feeding preference for AAs. In addition, different behavioural preferences have been detected between mated and virgin females¹³”.

As requested by the referee, we also tested the feeding preference for L-Gln and L-Ala in mated females. These novel data are shown in the **Supplemental Figure 3**. The following paragraph has been added in the Result section (lines 161-164):” The preference for 15 mM L-glutamine, L-alanine was also measured in mated female flies. Control females showed a clear preference for L-glutamine (PI = +0.4), while mutant females were indifferent to this AA. Moreover, control and mutant females showed similar PIs to L-alanine (**Supplementary Figure 3**)”.

7. Lines 213-214, what is the evidence for Obp-IR interaction? There was no text in the results which backed up this claim.

Our answer: We agree with the referee that our results do not support the Obp-IR interaction. To clarify this point we deleted the words “our data suggest” in the corresponding sentence in the Discussion section and now we write (lines 235-238): “In addition to its role in AA transport, and similarly to the LUSH OBP, OBP19b may also participate in the interaction with the protein receptor(s) involved in AA detection”.

8. Line 216, it might help your argument to add in the reference to Obp28a's unexpected buffering role (Larter et al.).

Our answer: As suggested by the referee, and to support the unexpected role of OBPs we added the “Larter et al., 2016” reference. The sentence was modified as follows (lines 235-240): “In addition to its role in AA transport, and similarly to the LUSH OBP, OBP19b may also participate in the interaction with the protein receptor(s) involved in AA detection. Similarly unexpected roles of OBPs were also recently described for OBP28a and OBP59a, which are crucial for odorant buffering and humidity detection, respectively^{37,38}”.

9. Line 217, could this be an artifact of in vitro expression?

Our answer: We believe that this is not an artefact of in vitro expression. Our main argument is the stereospecificity of binding observed with L-phenylalanine and L-glutamine which have been validated by behavioral and electrophysiological experiments.

10. Line 227, "GRN" instead of "GNR"

Our answer: We corrected this mistake.

11. Line 220, "OBP" instead of "OPB"

Our answer: We corrected this mistake.

12. Line 235, other Obps could also detect AAs, but you haven't ruled that out with the experiments presented here.

Our answer: We agree with the referee's remark that we cannot rule out the possibility that other Obps are involved in AA detection. To clarify this point, the following sentence was added (line 251) "However, we cannot rule out the possibility that other OBPs are involved in AA detection."

13. Line 250, what about Obp19b in the antenna? What could its role be in other organs?

Our answer: The role of OBP19b in the olfactory perception remains to be elucidated. We cannot rule out that this OBP binds also some odorants. To clarify this point, we added in the Discussion section the following sentence (lines 274-276): "The fact that OBP19b is also expressed—but at a much lower level—in olfactory appendages suggests that it could be also involved in odorant detection, but this remains to be elucidated."

-- Dr. Jennifer S. Sun

Reviewer #2 (Remarks to the Author):

In this study, Drosophila OBP19b was shown to bind to some amino acids, and a mutant for Obp19b gene was defective to show preference to these amino acids in feeding choice assay. Furthermore, and most surprisingly, these amino acids were shown to activate bitter-sensing neurons and OBP19b was necessary for it. These results seem inconsistent within themselves, but (although it may sound paradoxically) that makes this study so original and important that the finding should be shared by others immediately, which may stimulate further studies on this unexpected fact.

I think the experiments were performed more than satisfactorily, considering the wide range of approach from in vitro to in vivo assays, which is at the highest standard in the recent OBP studies. Nevertheless, the presentation of their results and conclusions are not perfect, in my opinion, as listed below.

1. The results shown in Fig.2b, Fig.2c, and Fig.2d are not very consistent to each other. In particular, why L-serine displaced NPN so little in Fig.2b while it did so much in Fig.2c? Any technical reasons?

Our answer: We agree with the referee's remark that our data do not look very consistent to each other, particularly for L-serine. This difference is due to technical reasons. The data presented in **Figure. 2a, b** were obtained with a Flexstation microplate reader whereas those presented in **Figure. 2c** were obtained with a Victor3 V microplate reader. To clarify this point, we performed the competitive screening assay presented in **Figure 2a, b** using the Victor3 V microplate. These novel results are now shown in the new **Figure 2**.

2. Related to the above comment, number of replicates was not shown in the all experiments except Table 1. What does the error bar indicate in each figure? It concerns how your results are reproducible.

Our answer: To cope with the referee's remark, we added the following sentence in the legends of **Figure 2** (lines 650-651) and **Supplementary figure 2** (lines 36-37): "Data values represent the mean \pm SEM based on three measurements obtained from three independent experiments."

3. In the binding assay shown in Fig.2b and d, what was the volume (or concentration) of the stock solutions of ligands?

Our answer: To clarify this point, we rewrote the following sentence in the legend of **Figure 2** (line 637-640): "The fluorescent displacement of the fluorescent probe NPN by various taste compounds was evaluated by adding aliquots of 50 mM ligand (dissolved in 50 mM sodium phosphate buffer at pH 7.5) to final concentration ranging from 0 to 20 mM."

4. Obp19b[1] = Obp19b-Gal4, but it is not obvious to the most readers (who may think Obp19[1] is a classical EMS allele). Should be explicitly explained that Obp19b[1] is a gene-targeting mutant in which Obp19b ORF was replaced with Gal4. 'Obp19b[1]>UAS-GFP' makes no sense.

Our answer: We totally agree with the referee's remark. All the manuscript and figures have been modified accordingly.

5. In Fig.3c, I don't see the expression is not so much overlapping. Both GFP and mCherry didn't have any sub-localization signal? Then why it looks like this? It may be more convincing to say OBP19b is expressed in trichogen.

Our answer: As also requested by referee #1, we performed new immunohistochemistry experiments. Our novel pictures show a partial overlapping between *nompA*-GFP and mCherry indicating that OBP19b is expressed in some thecogen cells. The novel **Figure 3c**, and **Figure 3e** have been modified accordingly.

6. *nompA*-Gal4? It must be *nompA*-GFP otherwise the experiment shown in Fig.3c was impossible, but described as '*nompA*-Gal4>UAS-GFP' in Materials & Methods and Fig.3 legend.

Our answer: We agree with the referee's remark. To clarify this point, we changed the following sentence in the Materials & Methods section (lines 306-307) : "The *nompA*-Gal4 line used to generate *nompA*-Gal4>UAS-GFP line was kindly provided by Seok Jun Moon and Craig Montell²⁸".

7. For the rescue experiment shown in Fig.4d, it is better to show the exact genotype of the rescued fly instead of Obp19b[1]>UAS-Obp19b. Unlike an usual crossing scheme for Gal4/UAS expression system, Obp19b-Gal4 needs to be homozygous in this experiment (UAS-Obp19b can be heterozygous). An explicit description is preferable as well as that for Obp19b[1] = Obp19b-Gal4.

Our answer: Now we explained how the cross was performed between transgenic strains to obtain the rescue genotype in the Materials & Methods section (Lines 303-306): "For the rescue, OBP19b-Gal4 homozygous females were crossed with UAS-OBP19b homozygous males to yield double heterozygous OBP19b-Gal4/+; UAS-OBP19b/+ F1 flies (OBP19b-Gal4>UAS-OBP19b)".

8. Results from electrophysiological analysis are really confusing. But I dare to say this is great because it does not look 'beautiful'. Nevertheless, the finding is so surprising from our current understanding in taste perception that some more discussion needs to be presented. Do you want to deny the 'labeled-line' model for taste perception with this finding? I suppose not. Then, readers may reasonably expect more thorough description about what kind of mechanisms can explain these self-inconsistent results, which should be supported by known facts. Or, if there are any technical limitations in the experimental design, they should be presented honestly with discussions about how they could influence the results and interpretation of them. I think it does not compromise any of this study's value.

Our answer: We agree with the reviewer that this finding was puzzling. Since we did not perform more experiments related to this finding, we kept in our data the experiment previously shown but we toned down our interpretation (the related sentence “We also discovered that taste neurons responding to L-phenylalanine can respond to caffeine, a bitter substance with an aversive effect”. is now deleted from the Abstract) and we are also more careful in our interpretation in the Discussion where we replaced “reveals” by “suggest” (line 258) and added the following sentence (lines 266-267): “Given that these results are still preliminary, they need to be confirmed with more behavioural and electrophysiological experiments”.

Reviewer #3 (Remarks to the Author):

Rihani and co-workers present an original work on the role of OBP19b on AA detection. This subject is very interesting as OBPs form a heterogeneous family of proteins that has been originally described as binding odorant but since shown has having diverse roles. It has been extensively developed recently partly through the work of Carlson's group, notably from their study of 2016 in *Elife* (Larter & Al.)

While the study is globally well written and well conducted, some data could be enhanced and expanded to better support the author's conclusions. It holds true from the sensilla localization of OBP19b to the behavioural and electrophysiological characterization of the mutant impact on taste modality. Below are major and minor comments.

Major:

1) The author's results should be compared with the recent study from Park and collaborators in 2017 focused on AA detection in *Drosophila*. Indeed, in the Park study, Phe elicits mostly response from S5, S7 and S9 and poor one from S6. The rationale of focusing on s6 sensilla should thus be discussed.

Our answer: We agree that our data could be compared with those obtained by Park & Carlson 2017. This is why we included an explanation in the Discussion (lines 251-256): “Our data also support that AA sensing is mainly mediated by s sensilla³⁹. The comparison of control flies response to 1 mM KCl solutions and to AA solutions with or without tricholine citrate (TCC, used to inhibit water neuron activity) indicates that TCC also inhibits AA responses. This is why we cannot compare our s6 electrophysiological responses to L-phenylalanine with a previous study using TCC in the stimulating solution tested³⁹”.

2) Similarly, in the Park study, Trp elicits strong response in S6, and Rihani and collaborators observe in the current study that Trp seems to be effectively bounded by OBP19b (even if it must be at the threshold of significance (very similar results to L-Phe in Fig.2). The present work would thus gain in clarity by exploring the detection of additional AA such as L-Trp or L-Ala. L-Trp detection should be very similarly affected to L-Phe in OBP19b mutant (at both electrophysiological and behaviour level). Additionally, exploring the electrophysiological response to L-Ala would be interesting as the behaviour detection of L-Ala which binds to the OBP according to Fig 2.D is not affected by in the OBP19b mutant. If the authors cannot provide such data, it should at least be discussed and temper their conclusion about the specificity of OBP19b toward L-Phe detection.

Our answer: For the sake of clarity, and as requested by the referee, we performed electrophysiological recordings of control and OBP19b null mutant to 10 mM L-tryptophan (L-Trp) and L-alanine (L-Ala). Binding experiments revealed that unlike L-glutamine (L-Gln) and L-serine (L-Ser), L-Ala or L-Trp did not significantly displaced NPN from the OBP19b binding cavity. The OBP19b deletion affects the electrophysiological responses to L-Ser and L-Gln but not those to L-Trp and L-Ala. These novel data are now shown in a new **Supplementary Figure 5a**. The following paragraph has been added in the Result section (lines 194-198): “We also

compared the electrophysiological response of the s6 sensillum in control and mutant flies to four other AAs (**Supplementary Figure 5a, b**). While a significant difference was observed between genotypes toward 10 mM L-serine and L-glutamine (two AAs shown to significantly bind to OBP19b; **Figure 2c**), no difference was detected to 10 mM L-tryptophan or L-alanine (which was shown not to bind OBP19b)".

3) In the OBP19b binding assay, the rationale of using sugars at μM concentration but AAs in the mM range should be explained as both are considered as nutrients.

Our answer: We agree with the referee's remark that sugars and L-AAs are nutrients and should have been tested at similar millimolar range. We performed the competitive binding assay with these three sugars at 15 mM. Interestingly we found that the sugars induced different displacements on NPN. These novel data are added in the **Figure 2**. Moreover, both the Materials & Methods (line 393), and the Results (line 118) sections were modified accordingly. These interesting findings will be investigated with further experiments in the close future.

4) As it is, the authors provide insufficient data to clearly establish which sensilla express OBP19b. Is it stable on different flies? How many did they examine?

Our answer: We found a similar labelling between different flies. Five series of experiments were performed, each time with ten proboscis. The number of replicates was added in the legend of **Figure 3**.

5) The electrophysiological dose responses to L-Phe contrast a lot with previous study (Park 2017). The basal firing is very high at 3 mM (similarly as recordings with the ineffective behaviourally D-Phe) and increase only slightly at 10 mM. Recordings with lower concentration showing the basal rate of firing would help to demonstrate the specificity of the response. Could this high basal firing rate be the response of the water cell? The authors explain in the M&M section that the water cell in s6 may only be active in mutant, which indicates important difference in basal function of taste sensilla and make the comparison between WT and mutant difficult. It is also puzzling that the KO of OBP19b seems to reduce this basal firing rate with L-Phe recordings but not with D-Phe. The authors should therefore temper their conclusions as in the discussion; they state "Our behavioral and electrophysiological comparison of control and mutant flies validates the specific role of OBP19b in the detection of L- but not D-phenylalanine. Moreover, the genetic excision of the OBP coding sequence (in mutant flies) and the transgenic rescue of it validated the speculation that OBP19b is specifically involved in the detection of L-phenylalanine and L-glutamine but not the detection of L-alanine or D-phenylalanine."

Our answer: We have now moved the sentence explaining the potential difference between our results and those of Park et al from the Materials & Methods section to the Discussion section (lines 251-256) "Our data also support that AA sensing is mainly mediated by s6 sensilla³⁹. The comparison of control flies response to 1 mM KCl solutions and to AA solutions with or without tricholine citrate (TCC, used to inhibit water neuron activity) indicates that TCC also inhibits AA responses. This is why we cannot compare our s6 electrophysiological responses to L-phenylalanine with a previous study using TCC in the stimulating solution tested³⁹". We never thought that basal activity of water cell was different between mutant and wild-type flies. To clarify this point, which was maybe unclear in our previous manuscript, we modified the sentence in the Materials & Methods section (lines 443-448): "In both control and mutant flies the stimulation with 10 mM L-phenylalanine induced two types of spikes. The smaller-amplitude spikes observed with 10 mM L-phenylalanine stimulation were produced by water-responding GRNs (W) based on the similar amplitude to those induced by 1 mM KCl (**Supplementary Figure 5b**)". Moreover, the water spikes in control and OBP19b-Gal4 recordings are now indicated by dots in the **Figure 5** and the corresponding spikes recordings

are represented in the **supplementary figure 5**. To follow the reviewer suggestion, we toned down our interpretation in our novel version and we replaced « validates » by « indicates » and « validated the » by « supports our » (lines 245-250).

6) Line 231: "Taken together, these results indicate that bitter-sensing GRNs can send information leading to either appetitive or aversive behavior." They authors should be much more cautious in their conclusion as it seems very counter intuitive to imagine that if L-Phe is only detected by bitter sensing neurons then their activation can lead to opposite behaviour when compared to activation with bitter compounds. In order to present this statement; they should genetically eliminate bitter sensing neurons and observe that both L-Phe detection in s6 sensilla and behaviourally attraction is abolished.

Our answer: We agree with the reviewer that this finding was puzzling. Since we did not perform more experiments related to this finding, we kept in our data the experiment previously shown but we toned down our interpretation (the related sentence "We also discovered that taste neurons responding to L-phenylalanine can respond to caffeine, a bitter substance with an aversive effect" is now deleted from the Abstract) and we are also more careful in our interpretation in the Discussion where we replaced ""reveals" by "suggest" (see line 258) and added the following sentence (lines 266-267): "Given that these results are still preliminary, they need to be confirmed with more behavioural and electrophysiological experiments". We also provide an alternative effect of bitter substance on insect behaviour (lines 260-261) "However, in another insect, synephrine, a bitter substance was shown to induce a preference behaviour⁴⁰" (Ozaki et al., 2011).

7) The mixture of caffeine and L-Phe in tip recording is consistent with the fact that L-Phe and caffeine are probably detected by the same cell but is not a real demonstration. Indeed, they authors present only one example of recording and no real spike analysis such as PSTH or ISI. It is however consistent with the Park study of 2017 showing that sensilla housing a bitter responding cell genetically ablated could not detect any more Trp. The authors should be more cautious in their assessment.

Our answer: We agree with the reviewer that this finding is preliminary. This is why we did not carry a PSTH or a ISI analysis. However, and to make this result clearer, we added at the left of each trace (**Figure 5d**), the overlay of the spikes indicating that the mixture induced a higher number of spikes (of similar shape) compared to both substances separately tested .

We kept in our data the experiment previously shown but we toned down our interpretation (the related sentence "We also discovered that taste neurons responding to L-phenylalanine can respond to caffeine, a bitter substance with an aversive effect". is now deleted from the Abstract) and we are also more careful in our interpretation in the Discussion where we replaced ""reveals" by "suggest" (see line 258) and added the following sentence (lines 266-267): "Given that these results are still preliminary, they need to be confirmed with more behavioural and electrophysiological experiments".

8) Could the authors provide a reference showing that an approximately 75% percentage of identity indicates a highly conserved sequence? 75% seems rather average considering this comparison is limited to Drosophila species

Our answer: To cope with the referee's remark, we added in the Result section (line 214) a reference showing that an approximately 75% percentage of identity indicates a highly conserved sequence (ref#36; Rost Prot. Eng. 12(2): 85-94, 1999).

Minor points:

1) In the introduction, the recent work on OBP particularly of Carlson team having both published mainly on AA detection and OBP roles is not mentioned (Larker 2016, Park 2017 and Sun 2018). It would be interesting to expand the introduction on the role that OBP may play outside olfaction such as in humidity detection and not only mention part of them in the discussion.

Our answer: Following the referee's request, now in the Introduction we mention the role outside olfaction of OBP in humidity detection. The following sentence has been added (lines 78-79) : "Recently, the *Drosophila* OBP59a was found to play a role in hygrosensation".

2) The rationale of using either virgin or mated females in the CAFE assay should be described before the discussion to improve the readability of the manuscript.

Our answer: To improve the readability of the manuscript, and according to the referee's remark, we added two sentences in the Results section (lines 153-154): "Previous studies have shown sexually dimorphic feeding preference for AAs. In addition, different behavioral preferences between mated and virgin females have been observed¹³".

3) line 26 " ionotropic receptor-expressing gustatory receptor neurons". Could you improve the readability of this part?

Our answer: We rewrote the sentence as follows (lines 25-27): "Mammals use G protein-coupled receptors to detect AAs, while insects such as the vinegar fly, *Drosophila melanogaster*, use gustatory neurons expressing ionotropic receptors".

4) On figure 3e: The sensilla expressing OBP19b should be annotated to link them better with the confocal images. Maybe using colour?

Our answer: According to the referee's remark, the sensilla expressing OBP19b are now indicated in magenta colour in **Figure 3d**. The figure legend has been modified accordingly.

5) Line 47: Precise the model "The general control nonderepressible 2 pathway detects the absence of AAs, whereas the Target of Rapamycin kinase pathway monitors particular AAs"

Our answer: The sentence has been modified as requested (lines 50-52): "In eukaryotes, the general control non-derepressible 2 pathway detects the absence of AAs, whereas the Target of Rapamycin kinase pathway process particular AAs^{3,4}".

6) Line 54 : The absence of umami taste is puzzling when speaking of AA detection in mammals. "Many AAs attract rodents and elicit a savory or sweet taste response in humans"

Our answer: The sentence has been modified as follows (lines 57-58): "Many AAs attract rodents and elicit a umami or sweet taste response in humans¹⁰".

7) Line 211: "In this study, we found that OBP19b is involved in the detection of hydrophilic molecules such as AAs". It seems a bit strange to qualify AA such as L-Phe of hydrophilic molecule. In fact, the authors should discuss how OBP19d could bind similarly Glu and Phe which are polar and hydrophobic respectively.

Our answer: According to the referee's comment, we changed the Discussion section as follows (lines 231-234): "In this study, we found that OBP19b is involved in the detection of hydrophilic AAs such as L-serine and L-glutamine, and hydrophobic AAs such as L-phenylalanine. Structural studies of OBP19b-AA complexes would help to understand the molecular determinants of AA binding to OBP19b".

8) Some part of the discussion could be shortened as they are quite speculative, notably the 3 paragraphs line 234, 241, 250. It would rather be interesting to expand the discussion on how their study fit in the model of OBP function. Is it to improve AA solubility? Similarly, it could be interesting to compare their results according to recent works on the roles and distribution of OBPs in *Drosophila* organs. For example, in the study from Larker 2016, the authors observed that OBP19b is not expressed in OSN similarly as the present study.

Our answer: According to referee's remark, we expanded the Discussion as follows (lines 235-240): "The AA transport role of OBP19b corresponds to the classical function attributed to OBPs. In addition to its role in AA transport, and similarly to the LUSH OBP, OBP19b may also participate in the interaction with the protein receptor(s) involved in AA detection. Similarly unexpected roles of OBPs were also recently described for OBP28a and OBP59a, which are crucial for odorant buffering and humidity detection, respectively^{37,38}".

Moreover and according to referee's remark, we added the following sentence (lines 228-231) "The restricted expression of OBP19b in gustatory appendages led to the suggestion that it had a role in taste detection. The low level of OBP19b expression in olfactory appendages is in agreement with the expression levels previously reported³⁷".

Reviewers' comments:

Reviewer #1 (Remarks to the Author):

This version of the manuscript is much improved. The findings are clearly presented and the data are more convincing.

Minor suggestion:

1. Line 79, please add the reference to Sun et al. 2018 at the end of the sentence on hygrosensation.

Reviewer #2 (Remarks to the Author):

The manuscript was revised appropriately except for one small, but important point that may influence the conclusion.

6. *nompA-Gal4*? It must be *nompA-GFP* otherwise the experiment shown in Fig.3c was impossible, but described as '*nompA-Gal4>UAS-GFP*' in Materials & Methods and Fig.3 legend.

Our answer: We agree with the referee's remark. To clarify this point, we changed the following sentence in the Materials & Methods section (lines 306-307) : "The *nompA-Gal4* line used to generate *nompA-Gal4>UAS-GFP* line was kindly provided by Seok Jun Moon and Craig Montell²⁸".

If this answer is true, the experimental design is flawed. The genotype of the fly will be *nompA-Gal4>UAS-GFP; OBP19b-Gal4>UAS-mCherry*, where GFP and mCherry are always co-expressed in the same pattern. Therefore, the result shown in Fig.3C has nothing to say about the co-expression of *Obp19b* and *nompA*. This must influence the conclusion substantially.

But the result shown in Fig.3C (not overlapping even in the replaced version) will never be obtained from the expected genotype, and I am still skeptical about the described method.

Reviewer #3 (Remarks to the Author):

In the present version of the manuscript, Karen Rihani and her colleagues have answered all of my concerns except for the electrophysiological part.

Indeed, as I requested more precision on spike sorting, the new version brings new information on the electrophysiological recordings. However, I find the spike sorting presentation still confusing for several reasons:

On Fig 5d, the cumulative spikes presented on the right should present all spikes sorted, not only those in the interval of 0.3 / 0.5 mV. One expects here only 2 sorts of spikes according to the author's interpretation, those <0.3 mV from the water cell and those > 0.3 mV from the L2 cells elicited by both caffeine and L-Phe. It would support their ability to sort spikes to see two clearly distinct spike classes. If there is a continuum of spikes amplitude with no clear cut off under 0.3 mV then there is no point of using this threshold. A critical point not mentioned by the authors is that spikes amplitude is very variable in *Drosophila* GRNs as it can change with different solution when recording the same sensilla on the same animal (see Fujishiro et al 1984). Thus, one can expect the mutation to simply change the spike amplitude and thus mislead the author's interpretations. If the two classes of spikes can be clearly separated with all substances used, then the interpretation would be much stronger.

On Fig 5b, the dots are now pointing the spikes from the cell responding to AA but in the previous version, it was from the water cell. If you compare both figures incoming from the same recording, many spikes are not sorted in those two categories. It should not be possible according to the spike sorting described in the M&M.

On Supp Fig 5, there is no spike sorting indicated and the recordings are really noisy making the interpretation quite difficult.

For all these reasons, I believe that all the spike sorting in the results and the interpretation linked to it should be simply removed from the manuscript. Furthermore, a sentence should indicate in the text that the electrophysiological interpretation could be biased due to an impact on water cell activity in the OBP19b mutant (most logically on Line 256 after the statement on TCC inhibition of AA response). It is particularly important here as TCC (usually used to inhibit water cell activity) inhibits the response to AA in the author's experiments.

Minor: There is no ref on line 79 concerning OBP59a role in hygrosensation.

Reviewers' comments and our reply:

Reviewer #1 (Remarks to the Author):

This version of the manuscript is much improved. The findings are clearly presented and the data are more convincing.

Minor suggestion:

1. Line 79, please add the reference to Sun et al. 2018 at the end of the sentence on hygrosensation.

Our reply: This reference is now included in the manuscript.

Reviewer #2 (Remarks to the Author):

The manuscript was revised appropriately except for one small, but important point that may influence the conclusion.

6. *nompA-Gal4*? It must be *nompA-GFP* otherwise the experiment shown in Fig.3c was impossible, but described as '*nompA-Gal4>UAS-GFP*' in Materials & Methods and Fig.3 legend.

*Our answer: We agree with the referee's remark. To clarify this point, we changed the following sentence in the Materials & Methods section (lines 306-307) : "The *nompA-Gal4* line used to generate *nompA-Gal4>UAS-GFP* line was kindly provided by Seok Jun Moon and Craig Montell28".*

If this answer is true, the experimental design is flawed. The genotype of the fly will be *nompA-Gal4>UAS-GFP;OBP19b-Gal4>UAS-mCherry*, where GFP and mCherry are always co-expressed in the same pattern. Therefore, the result shown in Fig.3C has nothing to say about the co-expression of Obp19b and *nompA*. This must influence the conclusion substantially.

But the result shown in Fig.3C (not overlapping even in the replaced version) will never be obtained from the expected genotype, and I am still skeptical about the described method.

Our reply: We made a mistake in the text and it is true that, as mentioned by the reviewer, we used the *nompA-GFP* direct fusion together with the combined *OBP19b-Gal4>UAS-mCherry* transgenes. We are sorry having made this mistake.

Indeed, the reviewer is right: with the two *nompA-Gal4* and *OBP19b-Gal4* drivers activating both *UAS-GFP* and *UAS-mCherry* fluorescent reporters, we would have obtained a complete overlap of both green and magenta colours (=white) due to the Gal4 induced by both drivers. Now, we have fixed this mistake and on our merged picture, we highlight with arrows the cells showing the clearest overlap. Note that the white color clearly appears when increasing the magnification of the picture. Not all cells show such a clear overlap maybe due to some difference in orientation.

Reviewer #3 (Remarks to the Author):

In the present version of the manuscript, Karen Rihani and her colleagues have answered all of my concerns except for the electrophysiological part.

Indeed, as I requested more precision on spike sorting, the new version brings new information on the electrophysiological recordings. However, I find the spike sorting presentation still confusing for several reasons:

On Fig 5d, the cumulative spikes presented on the right should present all spikes sorted, not only those in the interval of 0.3 / 0.5 mV. One expects here only 2 sorts of spikes according to the author's interpretation, those <0.3 mV from the water cell and those > 0.3 mV from the L2 cells elicited by both caffeine and L-Phe. It would support their ability to sort spikes to see two clearly distinct spike classes. If there is a continuum of spikes amplitude with no clear cut off under 0.3 mV then there is no point of using this threshold. A critical point not mentioned by the authors is that spikes amplitude is very variable in *Drosophila* GRNs as it can change with different solution when recording the same sensilla on the same animal (see Fujishiro et al 1984). Thus, one can expect the mutation to simply change the spike amplitude and thus mislead the author's interpretations. If the two classes of spikes can be clearly separated with all substances used, then the interpretation would be much stronger.

Our reply: The simplest way to deal with the reviewer comment was to eliminate this part of the electrophysiological experiment (corresponding to Fig. 5d). This does not change the scope of our paper given that this experiment is not necessary to follow the story on the role of the OBP19b in amino acid detection. We will perform additional experiments to confirm this preliminary finding.

On Fig 5b, the dots are now pointing the spikes from the cell responding to AA but in the previous version, it was from the water cell. If you compare both figures incoming from the same recording, many spikes are not sorted in those two categories. It should not be possible according to the spike sorting described in the M&M. On Supp Fig 5, there is no spike sorting indicated and the recordings are really noisy making the interpretation quite difficult. For all these reasons, I believe that all the spike sorting in the results and the interpretation linked to it should be simply removed from the manuscript. Furthermore, a sentence should indicate in the text that the electrophysiological interpretation could be biased due to an impact on water cell activity in the OBP19b mutant (most logically on Line 256 after the statement on TCC inhibition of AA response). It is particularly important here as TCC (usually used to inhibit water cell activity) inhibits the response to AA in the author's experiments.

Our reply: We have followed the reviewer advice and deleted the experiment (caffeine + L-phenylalanine; Fig.5d), and recordings (in the Suppl Fig.5b) to only keep the most important aspects of the electrophysiological experiments. We agree that our spike sorting was not satisfactory. In this revised manuscript we counted all the spikes without discriminating two kinds of spikes. The frequency of water spikes does not change between the two genotypes. We added the frequency of water spikes both in wild-type and mutant flies. Based on these revised data we can conclude that the mutant flies show an impaired response to L-Phe, supporting our behavioral results.

Minor: There is no ref on line 79 concerning OBP59a role in hygrometry.

Our reply: This reference is now included in the manuscript.

REVIEWERS' COMMENTS:

Reviewer #2 (Remarks to the Author):

My concerns were addressed appropriately.

Reviewer #3 (Remarks to the Author):

In the present version of the manuscript, Karen Rihani and her colleagues have answered my remaining concern about the accuracy of their spike sorting methodology.

Minor: Line 446, there is still one reference to the sorting of spikes: "In both control and mutant flies the stimulation with 10 mM L-phenylalanine induced two types of spikes." As the authors admitted that their sorting was really difficult, this should be removed.

On a side note for further studies of this group, the supplementary figure 6 displays a recording with a cell firing at a very regular rate and with a spike amplitude rising over time which is a classic artefact of tip recording. It clearly indicates that only one cell is active. Given the fact that the water cell should be active at such low concentration of L-Phe and its inhibition by TCC, it rather indicates that L-Phe is in fact activating the water cell.